# Automatic mapping of multiplexed social receptive fields by deep learning and GPU-accelerated 3D videography

Christian L. Ebbesen [1,2,3,4,5✉] & Robert C. Froemke [1,2,3,4,5✉]

Social interactions powerfully impact the brain and the body, but high-resolution descriptions of these important physical interactions and their neural correlates are lacking. Currently, most studies rely on labor-intensive methods such as manual annotation. Scalable and objective tracking methods are required to understand the neural circuits underlying social behavior. Here we describe a hardware/software system and analysis pipeline that combines 3D videography, deep learning, physical modeling, and GPU-accelerated robust optimization, with automatic analysis of neuronal receptive fields recorded in interacting mice. Our system ("3DDD Social Mouse Tracker") is capable of fully automatic multi-animal tracking with minimal errors (including in complete darkness) during complex, spontaneous social encounters, together with simultaneous electrophysiological recordings. We capture posture dynamics of multiple unmarked mice with high spatiotemporal precision (~2 mm, 60 frames/ s). A statistical model that relates 3D behavior and neural activity reveals multiplexed 'social receptive fields' of neurons in barrel cortex. Our approach could be broadly useful for neurobehavioral studies of multiple animals interacting in complex low-light environments.

[1] Skirball Institute of Biomolecular Medicine, New York University School of Medicine, New York, NY 10016, USA. [2] Neuroscience Institute, New York University School of Medicine, New York, NY 10016, USA. [3] Department of Otolaryngology, New York University School of Medicine, New York, NY 10016, USA. [4] Department of Neuroscience and Physiology, New York University School of Medicine, New York, NY 10016, USA. [5] Center for Neural Science, New York University, New York, NY 10003, USA. ✉email: christian.ebbesen@nyumc.org; robert.froemke@med.nyu.edu

Objective quantification of natural social interactions is difficult. The majority of our knowledge about rodent social behavior comes from hand-annotation of videos, yielding ethograms of discrete social behaviors such as 'social following', 'mounting', or 'anogenital sniffing'[1]. It is widely appreciated that these methods are susceptible to experimenter bias and have limited throughput. There is an additional problem with these approaches, in that manual annotation of behavior yields limited information about movement kinematics and physical body postures. This shortcoming is especially critical for studies relating neural activity patterns or other physiological signals to social behavior. For example, neural activity in many areas of the cerebral cortex is strongly modulated by movement and posture[2,3], and activity profiles in somatosensory regions can be difficult to analyze without understanding the physics and high-resolution dynamics of touch. Important aspects of social behavior, from gestures to light touch and momentary glances can be transient and challenging to observe in most settings, but critical to capturing the details and changes to social relationships and networks[4,5].

The use of deep convolutional networks to recognize objects in images has revolutionized computer vision, and consequently, also led to major advances in behavioral analysis. Drawing upon these methodological advances, several recent publications have developed algorithms for single animal[6–13] and multi-animal tracking[14–21]. These methods function by detection of key-points in 2D videos, and estimation of 3D postures is not straightforward in interacting animals, where some form of spatiotemporal regularization is needed to ensure that tracking is stable and error-free, even when multiple animals are closely interacting. During mounting or allo-grooming, for example, interacting animals block each other from the camera view, and tracking algorithms can fail. Having a large number of cameras film the animals from all sides can solve these problems[22,23], but this has required extensive financial resources for equipment, laboratory space, and processing power, which renders widespread use infeasible.

Some recent single[24]- and multi-animal[17–19] tracking methods have bypassed the problem of estimating the 3D posture of closely interacting animals by training a classifier to replicate human labeling discrete behavioral categories, such as attack and mounting. This approach is very powerful for automatically generating ethograms; however, for relating neural data to behavior, the lack of detailed information about movement and posture kinematics of interacting animals can be a critical drawback. In essentially every brain region, neural activity is modulated by motor signals[25–28] and vestibular signals[2,3,29]. Thus, any observed differences in neural activity between behavioral categories may simply be related instead to differences in movements and postures made by the animals in those different categories. To reveal how neural circuits process body language, touch, and other social cues[21] during a social interaction, descriptions of neural coding must be able to account for these important but complex motor- and posture-related activity patterns or confounds.

In parallel with deep-learning-based tracking methods, some studies have used depth-cameras for animal tracking, by fitting a physical 3D body-model of the animal to 3D data[30–32]. These methods are powerful because they can explicitly model the 3D movement and poses of multiple animals, throughout the social interaction. However, due to technical limitations of depth imaging hardware (e.g., frame rate, resolution, motion blur), to date it has been possible only to extract partial posture information about small and fast-moving animals, such as lab mice. Consequently, when applied to mice, these methods are prone to tracking mistakes when interacting animals get close to each other and the tracking algorithms require continuous manual supervision to detect and correct errors. This severely restricts throughput, making tracking across long time scales infeasible.

Here we describe a system for multi-animal tracking and neuro-behavioral data analysis that combines ideal features from both approaches: The 3D Deep-learning, Depth-video Social Mouse Tracker ("3DDD Social Mouse Tracker", https://github.com/chrelli/3DDD_social_mouse_tracker/). Our method fuses physical modeling of depth data and deep learning-based analysis of synchronized color video to estimate 3D body postures, enabling us to reliably track multiple mice during naturalistic social interactions. Our method is fully automatic (i.e., quantitative, scalable, and free of experimenter bias), is based on inexpensive consumer cameras, and is implemented in Python, a simple and widely used computing language. Our method is capable of tracking the animals using only infrared video channels (i.e., in visual darkness for mice, a nocturnal species), is self-aligning, and requires only a few hundred labeled frames for training. We combine our tracking method with silicon probe recordings of single-unit activity in barrel cortex to demonstrate the usefulness of a continuous 3D posture estimation and an interpretable body model: We implement a full-automatic neural data analysis pipeline (included along with the tracking code), that yields a population-level map of neural tuning to the features of a social interaction (social touch, movements, postures, spatial location, etc.) directly from raw behavior video and spike trains.

## Results

**Raw data acquisition.** We built an experimental setup that allowed us to capture synchronized color images and depth images from multiple angles, while simultaneously recording synchronized neural data (Fig. 1a). We used inexpensive, state-of-the-art 'depth cameras' for computer vision and robotics. These cameras contain several imaging modules: one color sensor, two infrared sensors, and an infrared laser projector (Fig. 1b). Imaging data pipelines, as well as intrinsic and extrinsic sensor calibration parameters can be accessed over USB through a C/C++ SDK with Python bindings. We placed four depth cameras, as well as four synchronization LEDs around a transparent acrylic cylinder which served as our behavioral arena (Fig. 1c).

Each depth camera projects a static dot pattern across the imaged scene, adding texture in the infrared spectrum to reflective surfaces (Fig. 1d). By imaging this highly-textured surface simultaneously with two infrared sensors per depth camera, it is possible to estimate the distance of each pixel in the infrared image to the depth camera by stereopsis (by locally estimating the binocular disparity between the textured images). Since the dot pattern is static and only serves to add texture, multiple cameras do not interfere with each other and it is possible to image the same scene simultaneously from multiple angles. Simultaneous capture from all angles is one key aspect of our method, not possible with depth imaging systems that rely on actively modulated light (such as the Microsoft Kinect system and earlier versions of the Intel Realsense cameras, where multi-view capture requires offset capture times).

Since mouse movement is fast (on a millisecond time scale[33]), it is vital to minimize motion blur in the infrared images and thus the final 3D data ('point-cloud'). To this end, our method relies on two key features. First, we use depth cameras where the infrared sensors have a global shutter (e.g., Intel D435) rather than a rolling shutter (e.g., Intel D415). Using a global shutter reduces motion blur in individual image frames, but also enables synchronized image capture across cameras. Without synchronization between cameras, depth images are taken at different times, which adds blur to the composite point-cloud. We set

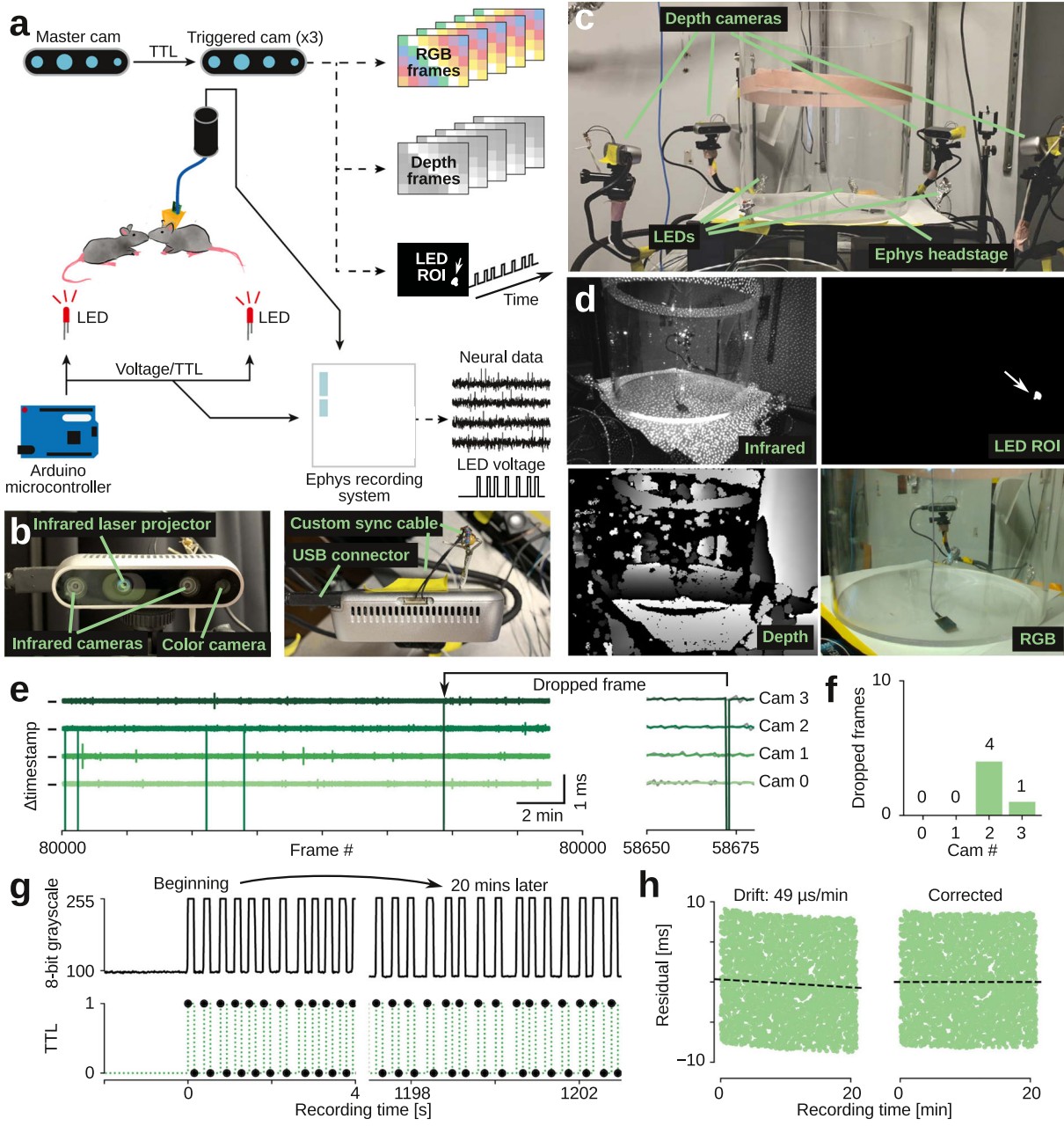

**Fig. 1 Raw data acquisition, temporal alignment, and recording stability. a** Schematic of recording setup, showing the flow of synchronization pulses and raw data. We use a custom Python program to record RGB images, depth images, and state (on/off) of synchronization LEDs from all four cameras. Neural data and TTL state of LEDs are recorded with a standard electrophysiology recording system. We use a custom Python program to record video frames over USB (60 frames/s) and automatically deliver LED synchronization pulses with randomized delays via Arduino microcontroller. **b** Close-up images of the depth cameras, showing the two infrared sensors, color sensor, and cables for data transfer and synchronization. **c** Photograph of recording setup, showing the four depth cameras, synchronization LEDs, and circular behavioral arena (transparent acrylic, 12" diameter). **d** Example raw data images (top left: single infrared image with visible infrared laser dots; top right: corresponding automatically-generated mask image for recording LED synchronization state (arrow, LED location); bottom left: corresponding depth image, estimated from binocular disparity between two infrared images; bottom right: corresponding color image). **e** Inter-frame-interval from four cameras (21 min of recording). Vertical ticks indicate 16.66 ms (corresponding to 60 frames/ s), individual cameras are colored and vertically offset. Frame rate is very stable (jitter across all cameras: ±26 μs). Arrow, example dropped frame. **f** Number of dropped frames across the example 21 min recording. **g** Top row, LED state (on/off) as captured by one camera (the 8-bit value of central pixel of LED ROI mask), at start of recording and after 20 min of recording. Bottom row, aligned LED trace, as recorded by electrophysiology recording system. **h** Temporal residuals between recorded camera LED trace (**g**, top) and recorded TTL LED trace (**g**, bottom) are stable, but drift slightly (49 μs/min, left panel). We can automatically detect and correct for this small drift (right panel). Source data are provided as a Source Data file.

custom firmware configurations in our recording program, such that all infrared sensors on all four cameras are hardware-synchronized to each other by TTL-pulses via custom-built, buffered synchronization cables (Fig. 1b).

We wrote a custom multithreaded Python program with online compression, that allowed us to capture the following types of raw data from all four cameras simultaneously: 8-bit RGB images (320 × 210 pixels, 60 frames/s), 16-bit depth images (320 × 240 pixels, 60 frames/s) and the 8-bit intensity trace of a blinking LED (60 samples/s, automatically extracted in real-time from the infrared images). Our program also captures camera meta-data, such as hardware time-stamps and frame numbers of each image, which allows us to identify and correct for possible dropped frames. On a standard desktop PC, the recording system had very few dropped frames and the video recording frame rate and the imaging and USB image transfer pipeline were stable (Fig. 1e, f).

**Temporal stability and temporal alignment.** In order to relate tracked behavioral data to neural recordings, we need precise temporal synchronization. Digital hardware clocks are generally stable but their internal speed can vary, introducing drift between clocks. Thus, even though all depth cameras provide hardware timestamps for each acquired image, for long-term recordings, across behavioral time scales (hours to days), a secondary synchronization method is required.

For synchronization to neural data, our recording program uses a USB-controlled Arduino microprocessor to output a train of randomly-spaced voltage pulses during recording. These voltage pulses serve as TTL triggers for our neural acquisition system (sampled at 30 kHz) and drive LEDs, which are filmed by the depth cameras (Fig. 1a). The cameras sample an automatically detected ROI to sample the LED state at 60 frames/s, integrating across a full infrared frame exposure (Fig. 1g). We use a combination of cross-correlation and robust regression to automatically estimate and correct for shift and drift between the depth camera hardware clocks and the neural data. Since we use random pulse trains for synchronization, alignment is unambiguous and we can achieve super-frame-rate-precision. In a typical experiment, we estimated that the depth camera time stamps drifted with ~49 μs/min. For each recording, we automatically estimate and correct for this drift to yield stable residuals between TTL flips and depth frame exposures (Fig. 1h). Note that the neural acquisition system is not required for synchronization, so for a purely behavioral study, we can run the same LED-based protocol to correct for potential shift and drift between cameras by choosing one camera as a reference.

**Detection of body key-points by deep learning.** We pre-processed the raw image data to extract two types of information for the tracking algorithm: the location in 3D in space of body key-points and the 3D point-cloud corresponding to the body surface of the animals. We used a deep convolutional neural network to detect key-points in the RGB images, and extracted the 3D point-cloud from the depth images (Fig. 2a). For key-point detection (nose, ears, base of tail, and neural implant for implanted animals), we used a 'stacked hourglass network'[34]. This type of encoder-decoder network architecture combines residuals across successive upsampling and downsampling steps to generate its output, and has been successfully applied to human pose estimation[34] and limb tracking in immobilized flies[35] (Fig. 2b, details of network architecture in Supplementary Fig. 1).

We used back-propagation to train the network to output four 'target maps', each indicating the pseudo-posterior probability of each type of key-point, given the input image. The target maps were generated by manually labeling the key-points in training frames, followed by down-sampling and convolution with Gaussian kernels (Fig. 2c, 'targets'). We selected the training frames using image clustering to avoid redundant training on very similar frames[8]. The manual key-point labeling can be done with any labeling software. We customized a version of the lightweight, open source labeling GUI from the 'DeepPoseKit' package[8] for the four types of key-points, which we provide as supplementary software (Supplementary Fig. 2).

In order to improve key-point detection, we used two additional strategies. First, we also trained the network to predict 'affinity fields'[36], which have been shown to improve human[36] and animal[8,15] body key-point tracking. We used '1D' affinity fields (as in ref. [8]) generated by convolving the path between labeled body key-points that are anatomically connected in the animal. With our four key-points, we added seven affinity fields (e.g., 'nose-to-ears', 'nose-to-tail'), that together form a skeletal representation of each body (Fig. 2c, 'affinity fields'). Thus, from three input channels (RGB pixels), the network predicts eleven output channels (Fig. 2d). As the stacked hourglass architecture involves intermediate prediction, which feeds back into subsequent hourglass blocks (repeated encoding and decoding, Fig. 2b), prediction of affinity fields feeds into downstream predictions of body key-points. This leads to the improvement of downstream key-point predictions, because the affinity fields give the network access to holistic information about the body. The intuitive probabilistic interpretation is that instead of simply asking questions about the keypoints (e.g., 'do these pixels look like an ear?'), we can increase predictive accuracy by considering the body context (e.g., 'these pixels sort of look like an ear, and those pixels sort of look like a nose—but does this path between the pixels also look like the path from an ear to a nose?').

The second optimization approach was image data augmentation during training[37]. Instead of only training the network on manually-labeled images, we also trained the network on morphed and distorted versions of the labeled images (Supplementary Fig. 3). Training the network on morphed images (e.g., rotated or enlarged), gives a similar effect to training on a much larger dataset of labeled images, because the network then learns to predict many artificially generated, slightly different views of the animals. Training the network on distorted images is thought to reduce overfitting on single pixels and reduce the effect of motion blur[37].

Using a training set of 526 images, and by automatically adjusting learning rate during training, the network was well-trained (plateaued) within one hour of training on a standard desktop computer (Fig. 2e), yielding good predictions of both body key-points and affinity fields (Fig. 2f).

**All-infrared tracking.** As mice are nocturnal, we also developed a version of the tracking software that only relies on the infrared video stream (i.e., in visual darkness for the mice). This facilitates the study of naturalistic social interactions in darkness. For 'all-infrared' experiments, the arena was lit with infrared LED lamps, and the software was changed to save only the infrared images (16-bit, 640 × 448, 60 frames/s). Detection of body key-points by deep learning from in these images are made difficult by the prominent infrared laser dot pattern (Fig. 2g). We trained the deep neural network to ignore the dot pattern by using a data augmentation strategy. We recorded and labeled body parts in a training data set (720 images), where the infrared laser was turned off, and trained the network on labeled images augmented with a probabilistically generated noise pattern of white dots with a similar size and density to the 'real' laser pattern (Fig. 2h). A network trained on these data allowed us to successfully detect body key-points in real images with the infrared laser turned on (Fig. 2i).

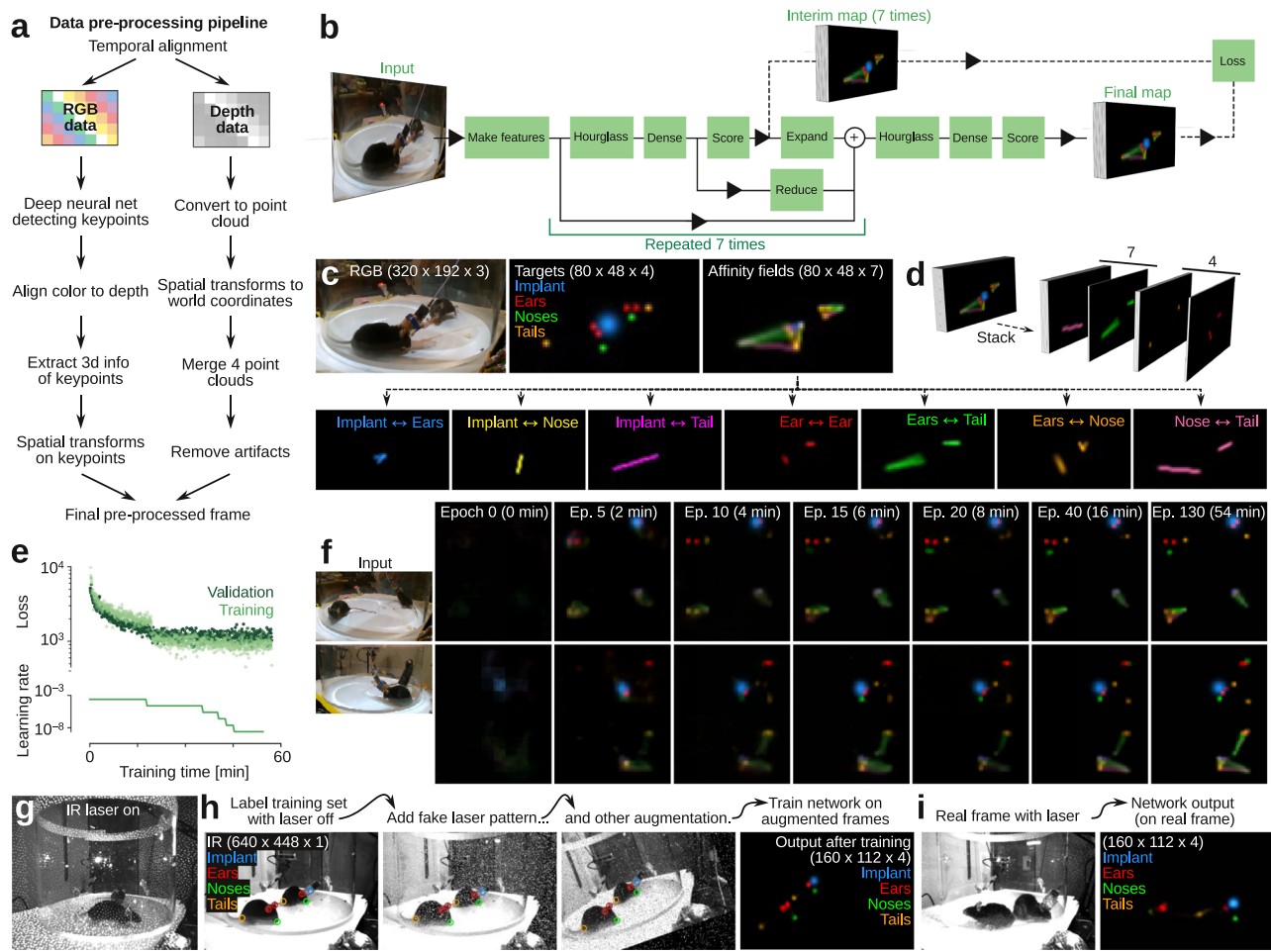

**Fig. 2 Detection of body key-points with a deep convolutional neural network. a** Workflow for pre-processing of raw image data. **b** The 'stacked hourglass' convolutional network architecture. Each 'hourglass' block of the network uses pooling and upsampling to incorporate both fine (high-resolution) and large-scale (low-resolution) information in the target prediction. The hourglass and scoring blocks are repeated seven times (seven 'stacks'), such that intermediate key-point and affinity field predictions feed into subsequent hourglass blocks. Both the intermediate and final target maps contribute to the training loss, but only the final output map is used for prediction. **c** Example training data for the deep convolutional neural network. The network is trained to output four types of body key-points (Implant, Ears, Noses, and Tails) and seven 1-D affinity fields, connecting key-points within each body. **d** Example of full training target tensor. **e** Convergence plot of example training set. Top, loss function for each mini-batch of the training set (526 images) and validation set (50 images). Bottom, learning rate. Network loss is trained (plateaued) after ~ 60 min. **f** Network performance as function of training epoch for two example images in the validation set. Left, input images; right, final output maps for key-points and affinity fields. **g** In an infrared frame (under infrared illumination), the clear view of the mice is 'obstructed' by the infrared laser dot pattern. **h** Example labeled training frame (with the laser turned off), showing the augmentation strategy of applying a probabilistically generated 'fake' laser dot pattern during training. **i** Example network output of the trained network on a 'real' infrared frame with the infrared laser turned on.

To optimize the network architecture and estimate pseudo-posterior probability cutoffs in the network output maps with a good tradeoff between missed body key-points, false positives, and network training/inference time, we profiled the network across the number of hourglass stacks (Supplementary Figs. 4, 5), with and without various types of training data augmentation (Supplementary Fig. 6), and with and without part affinity fields (Supplementary Fig. 7). Based on the hand-labeled validation data, we found that 3 hourglass stacks and a pseudo-posterior probability cutoff of 0.5 led to good performance (Supplementary Figs. 4–7).

**Pre-processing of 3D video**. By aligning the color images to the depth images, and aligning the depth images in 3D space, we could assign three-dimensional coordinates to the detected key-points. We pre-processed the depth data to accomplish two goals. First, we wanted to align the cameras to each other in space, so we could fuse their individual depth images to one single 3D point-cloud. Second, we wanted to extract only points corresponding to the animals' body surfaces from this composite point-cloud.

To align the cameras in space, we filmed the trajectory of a sphere that we moved around the behavioral arena. We then used a combination of motion filtering, color filtering, smoothing, and thresholding to detect the location of the sphere in the color frame, extracted the partial 3D surface from the aligned depth data, and used a robust regression method to estimate the center coordinate (Fig. 3a). This procedure yielded a 3D trajectory in the reference frame of each camera (Fig. 3b) that we could use to robustly estimate the transformation matrices needed to bring all trajectories into the same frame of reference (Fig. 3c). This robust alignment is a key aspect of our method, as errors can easily be introduced by moving the sphere too close to a depth camera or out of the field of view during recording (Fig. 3b, c, arrow). After alignment, the median camera-to-camera difference in the estimate of the center coordinate of the 40 mm-diameter sphere was only 2.6 mm across the entire behavioral arena (Fig. 3d, e).

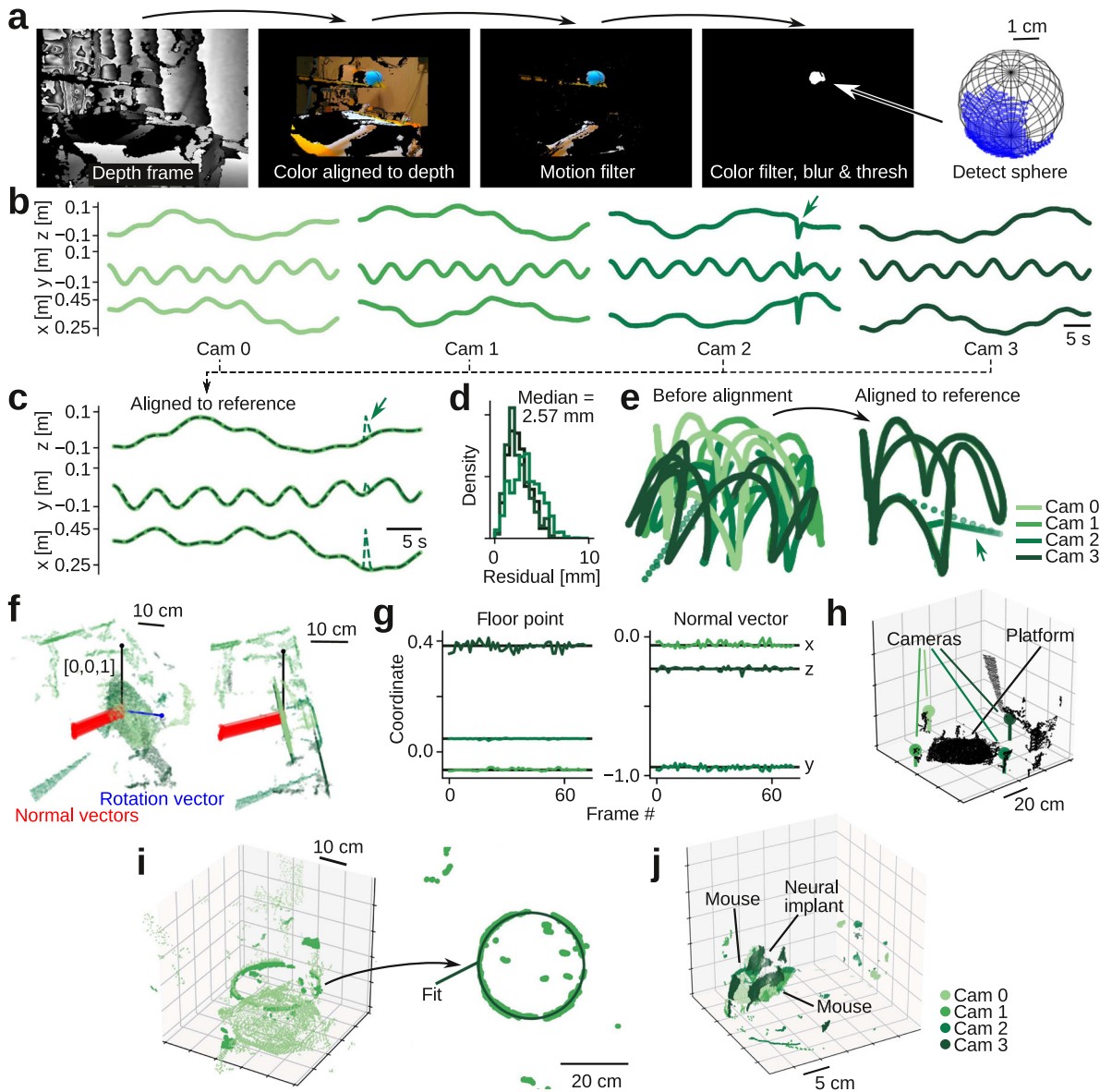

**Fig. 3 Depth data alignment and pre-processing. a** Calibration ball detection pipeline. We use a combination of motion filtering, color filtering, and smoothing filters to detect and extract 3D ball surface. We estimate 3D location of the ball by fitting a sphere to the extracted surface. **b** Estimated 3D trajectories of calibration ball as seen by the four cameras. One trajectory has an error (arrow) where ball trajectory was out of view. **c** Overlay of trajectories after alignment in time and space. Our alignment pipeline uses a robust regression method and is insensitive to errors (arrow) in the calibration ball trajectory. **d** Distribution of residuals, using cam 0 as reference. **e** Estimated trajectory in 3D space, before and after alignment of camera data. **f** Example frame used in automatic detection of the behavioral arena location. Show are pixels from the four cameras, after alignment (green), estimated normal vectors to the behavioral platform floor (red), the estimated rotation vector (blue), and the reference vector (unit vector along z-axis, black). **g** Estimated location (left) and normal vector (right) to the behavioral platform floor, across 60 random frames. **h** Example frame, after rotating the platform into the xy-plane, and removing pixels below and outside the arena. Inferred camera locations are indicated with stick and ball. **i** Automatic detection of behavioral arena location. **j** Example 3D frame, showing merged data from four cameras, after automatic removal of the arena floor and imaging artifacts induced by the acrylic cylinder. Colors, which camera captured the pixels.

We used a similar robust regression method to automatically detect the base of the behavioral arena. We detected planes in composite point-cloud (Fig. 3f) and used the location and normal vector, estimated across 60 random frames (Fig. 3g), to transform the point-cloud such that the base of the behavioral arena laid in the xy-plane (Fig. 3h). To remove imaging artifacts stemming from light reflection and refraction due to the curved acrylic walls, we automatically detected the location and radius of the acrylic cylinder (Fig. 3i). With the location of both the arena base and the acrylic walls, we used simple logic filtering to remove all points associated with the base and walls, leaving only points

inside the behavioral arena (Fig. 3j). Note that if there is no constraint on laboratory space, an elevated platform can be used as a behavioral arena, eliminating imaging artifacts associated with the acrylic cylinder.

**Loss function design**. The pre-processing pipeline described above takes color and depth images as inputs, and outputs two types of data: a point-cloud, corresponding to the surface of the two animals, and the 3D coordinates of detected body key-points (Fig. 4a and Supplementary Video 1). To track the body postures of interacting animals across space and time, we developed an

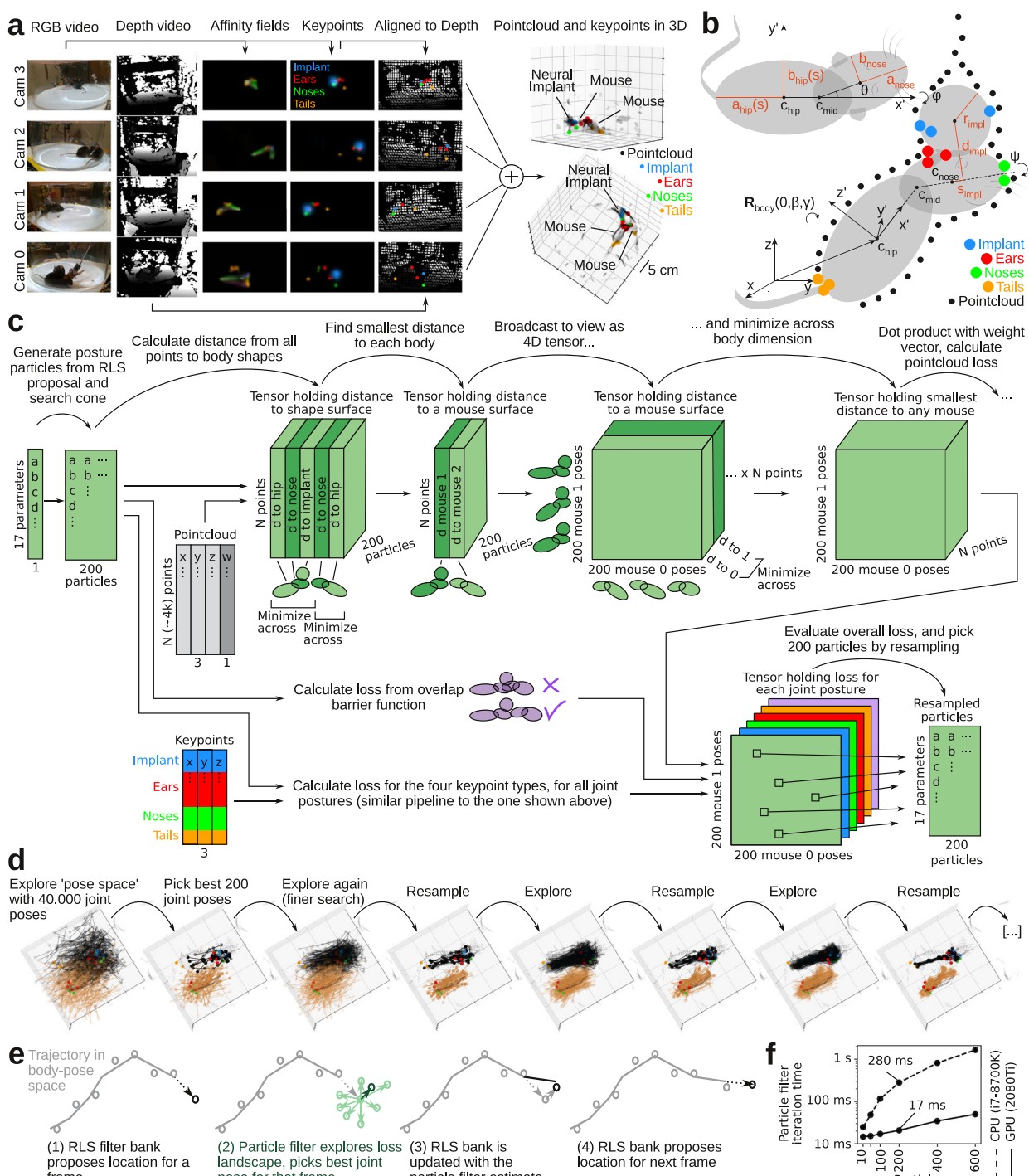

**Fig. 4 Mouse body model and GPU-accelerated tracking algorithm. a** Full assembly pipeline for a single pre-processed data frame, going from raw RGB and depth images (left columns) to assembled 3D point-cloud (black dots, right) and body key-point positions in 3D space (colored dots, right). **b** Schematic depiction of mouse body model (gray, deformable ellipsoids) and implant model (gray sphere), fit to point-cloud (black dots) and body key-points (colored dots). The loss function assigns loss to distance from the point-cloud to the body model surface (black arrows) and from key-point locations to landmark locations on the body model (e.g., from nose key-points to the tip of the nose ellipsoids; colored arrows). **c** Schematic of loss function calculation and tracking algorithm. All operations implemented as GPU-accelerated tensor algebra. **d** Example steps showing convergence of the particle filter on a single frame. **e** Schematic depiction of the two levels of the tracking algorithm: Within a single frame, the joint poses are estimated with the particle filter. Between frames, the RLS filter bank incorporates information from multiple previous frames to estimate and propose the minimum in 'pose space'. **f** Iteration time of a particle filter step, as a function of particles, on a GPU and CPU. For 200 particles (i.e., 40.000 joint poses), the GPU-accelerated particle filter is ~16.5 times faster than the CPU.

algorithm that incorporates information from both data types. The basic idea of the tracking algorithm is that for every frame, we fit the mouse bodies by minimizing a loss function of both the point-cloud and key-points, subject to a set of spatiotemporal regularizations.

For the loss function, we made a simple parametric model of the skeleton and body surface of a mouse. The body model consists of two prolate spheroids (the 'hip ellipsoid' and 'head ellipsoid'), with dimensions based on an average adult mouse (Fig. 4b). The head ellipsoid is rigid, but the hip ellipsoid has a free parameter ($s$) modifying the major and minor axes to allow the hip ellipsoids to be longer and narrower (e.g., during stretching, running, or rearing) or shorter and wider (e.g., when still or self-grooming). The two ellipsoids are connected by a joint that allows the head ellipsoid to turn left/right and up/down within a cone corresponding to the physical movement limits of the neck.

Keeping the number of degrees of freedom low is vital to make loss function minimization computationally feasible[38]. Due to the rotational symmetry of the ellipsoids, we could choose a parametrization with 8 degrees of freedom per mouse body: the central coordinate of the hip ellipsoid ($x, y, z$), the rotation of the major axis of the hip ellipsoid around the $y$- and $z$-axis ($\beta, \gamma$), the left/right and up/down rotation of the head ellipsoid ($\theta, \varphi$), and the stretch of the hip ellipsoids ($s$). For the implanted animal, we added an additional sphere to the body model, approximating the surface of the head-mounted neural implant (Fig. 4b). The sphere is rigidly attached to the head ellipsoid and has one degree of freedom; a rotational angle ($\psi$) that allows the sphere to rotate around the head ellipsoid, capturing head tilt of the implanted animal. Thus, in total, the joint pose (the body poses of both mice) was parametrized by only 17 variables.

To fit the body model, we adjusted these parameters to minimize a weighted sum of two loss terms: (i) The shortest distance from every point in the point-cloud to body model surface. (ii) The distance from detected key-points to their corresponding location on the body model surface (e.g., nose key-points near the tip of one of the head ellipsoids, tail key-points near the posterior end of a hip ellipsoid).

We then used several different approaches for optimizing the tracking. First, for each of the thousands of point in the point-cloud, we needed to calculate the shortest distance to the body model ellipsoids. Calculating these distances exactly is not computationally feasible, as this requires solving a six-degree polynomial for every point[39]. As an approximation, we instead used the shortest distance to the surface, along a path that passes through the centroid (Supplementary Fig. 8a, b). Calculating this distance could be implemented as pure tensor algebra[40], which could be executed efficiently on a GPU in parallel for all points simultaneously. Second, to reduce the effect of imaging artifacts in the color and depth imaging (which can affect both the point-cloud or the 3D coordinates of the key-points), we clipped distance losses at 3 cm, such that distant 'outliers' do contribute and not skew the fit (Supplementary Fig. 8c). Third, because pixel density in the depth images depends on the distance from the depth camera, we weighed the contribution of each point in the point-cloud by the squared distance to the depth camera (Supplementary Fig. 8d). Fourth, to ensure that the minimization does not converge to unphysical joint postures (e.g., where the mouse bodies are overlapping), we added a penalty term to the loss function if the body models overlap. Calculating overlap between two ellipsoids is computationally expensive[41], so we computed overlaps between implant sphere and spheres centered on the body ellipsoids with a radius equal to the minor axis (Supplementary Fig. 8f). Fifth, to ensure spatiotemporal continuity of body model estimates, we also added a penalty term to

the loss function, penalizing overlap between the mouse body in the current frame, and other mouse bodies in the previous frame. This ensures that the bodies do not switch place, something that could otherwise happen if the mice are in joint poses with certain mirror symmetries (Supplementary Fig. 8g, h).

**GPU-accelerated robust optimization.** Minimizing the loss function requires solving three major challenges. The first challenge is computational speed. The number of key-points and body parts is relatively low (~tens), but the number of points in the point-cloud is large (~thousands), which makes the loss function computationally expensive. For minimization, we need to evaluate the loss function multiple times per frame (at 60 frames/s). If loss function evaluation is not fast, tracking becomes unusably slow. The second challenge is that the minimizer has to properly explore the loss landscape within each frame and avoid local minima. In early stages of developing this algorithm, we were only tracking interacting mice with no head implant. In that case, for the small frame-to-frame changes in body posture, the loss function landscape was nonlinear, but approximately convex, so we could use a fast, derivative-based minimizer to track changes in body posture (geodesic Levenberg-Marquardt steps[38]). For use in neuroscience experiments, however, one or more mice might carry a neural implant for recording or stimulation. The implant is generally at a right angle and offset from the 'hinge' between the two hip and head ellipsoids, which makes the loss function highly non-convex[42]. The final challenge is robustness against local minima in state space. Even though a body posture minimizes the loss in a single frame, it might not be an optimal fit, given the context of other frames (e.g., spatiotemporal continuity, no unphysical movement of the bodies).

To solve these three challenges—speed, state space exploration, and spatiotemporal robustness—we designed a custom GPU-accelerated minimization algorithm, which incorporates ideas from annealed particle filters[43] and online Bayesian filtering (Fig. 4c). To maximize computational speed, the algorithm was implemented as pure tensor algebra in Pytorch, a high-performance GPU computing library[44]. Annealed particle filters are suited to explore highly non-convex loss surfaces[43], which allowed us to avoid local minima within each frame. Between frames, we used online filtering, to avoid being trapped in low-probability solutions given the context of the preceding tracking. For every frame, we first proposed the state of the 17-parameters using a recursive least-squares ('RLS') filter bank trained on preceding frames. After particle filter-based loss function minimization within a single frame, we updated the RLS filter bank, and proposed a particle filter starting point for the next frame (Fig. 4d, e).

The 'two-layer' tracking strategy (particle filter within frames and RLS filter between frames) has three major advantages. First, by proposing a solution from the RLS bank, we often already start the loss function minimization close to the new minimum. Second, if the RLS filter deems that the fit for a single frame is unlikely (an outlier), based on the preceding frames, this fit will only weakly update the filter bank, and thus only weakly perturb the upcoming tracking. This gives us a convenient way to balance the information provided by the fit of a single frame, and the 'context' provided by previous frames. Third, the RLS filter-based approach is only dependent on previously tracked frames, not future frames. This is in contrast to other approaches to incorporating context that rely on versions of backwards belief propagation[5,16,35]. Note that since our algorithm only relies on past data for tracking, it is possible—in future work—to optimize our algorithm for real-time use in closed-loop experiments.

For each recording, we first automatically initiated the tracking algorithm: We automatically scanned forward in the video to find

a frame, where the mice were well separated (assessed by $k$-means clustering of the 3D positions of the body key-points into two clusters, and by requiring that the 'cross-mouse' cluster distance is at least 5 cm (Supplementary Fig. 9). From this starting point, we explored the loss surface with 200 particles (Fig. 4d). We generated the particles by perturbing the proposed minimum by quasi-random, low-discrepancy sampling[45] (Supplementary Fig. 10). We exploited the fact that the loss function structure allowed us to execute several key steps in parallel, across multiple independent dimensions, and implemented these calculations as vectorizes tensor operations. This allowed us to leverage the power of CUDA kernels for fast tensor algebra on the GPU[44]. Specifically, to efficiently calculate the point-cloud loss (shortest distance from a point in the point-cloud to the surface of a body model), we calculated the distance to all five body model spheroids for all points in the point-cloud and for all 200 particles, in parallel (Fig. 4c). We then applied fast minimization kernels across the two body models, to generate a smallest distance to either mouse, for all points in the point cloud. Because the mouse body models are independent, we only had to apply a minimization kernel to calculate the smallest distance, for every point, to 40,000 (200 × 200) joint poses if the two mice. These parallel computation steps are a key aspect of our method, which allows our tracking algorithm to avoid the 'curse of dimensionality', by not exploring a 17-dimensional space, but rather explore the intersection of two independent 8-dim and 9-dim subspaces in parallel. We found that our GPU-accelerated implementation of the filter increased the processing time of a single frame by more than an order of magnitude compared to a fast CPU (e.g., ~16-fold speed increase for 200 particles, Fig. 4f).

**Tracking algorithm performance**. To ensure that the tracking algorithm did not get stuck in suboptimal solutions, we forced the particle filter to explore a large search space within every frame (Supplementary Fig. 11a–c). In successive iterations, we gradually made perturbations to the particles smaller and smaller by annealing the filter[43]), to approach the minimum. At the end of each iteration, we 'resampled' the particles by picking the 200 joint poses with the lowest losses in the 200-by-200 matrix of losses. This 'top-k' resampling strategy has two advantages. First, it can be done without fully sorting the matrix[46], the most computationally expensive step in resampling[47]. Second, it provides a type of 'importance sampling'. During resampling, some poses in the next iteration might be duplicates (picked from the same row or column in the 200-by-200 loss matrix.), allowing particles in each subspace to collapse at different rates (if the particle filter is very certain about one body pose, but not the other, for example).

By investigating the performance of the particle filter across iterations, we found that the filter generally converged sufficiently within five iterations (Supplementary Fig. 11d, Supplementary Video 2) to provide good tracking across frames (Supplementary Fig. 11e). In every frame, the particle filter fit yields a noisy estimate of the 3D location of the mouse bodies. The transformation from the joint pose parameters (e.g., rotation angles, spine scaling) to 3D space is highly nonlinear, so simple smoothing of the trajectory in pose parameter space would distort the trajectory in real space. Thus, we filtered the tracked trajectories by a combination of Kalman-filtering and maximum likelihood-based smoothing[48,49] and 3D rotation smoothing in quaternion space[50] (Supplementary Fig. 12 and Supplementary Video 3).

Representing the joint postures of the two animals with this parametrization was highly data efficient, reducing the memory footprint from ~3.7 GB/min for raw color/depth image data, to ~0.11 GB/min for pre-processed point-cloud/key-point data to ~1 MB/min for tracked body model parameters. On a regular desktop computer with a single GPU, we could do key-point detection in color image data from all four cameras in ~2× real time (i.e., it took 30 min to process a 1 h experimental session). Depth data processing (point-cloud merging and key-point deprojection) ran at ~0.7× real time, and the tracking algorithm ran at ~0.2× real time (if the filter uses 200 particles and 5 filter iterations per frame). Thus, for a typical experimental session (~ hours), we would run the tracking algorithm overnight, which is possible because the code is fully automatic.

**Error detection**. Error detection and correction is a critical component of behavioral tracking. Even if error rates are nominally low, errors are non-random, and errors often happen exactly during the behaviors in which we are most interested: interactions. In multi-animal tracking, two types of tracking error are particularly fatal as they compound over time: identity errors and body orientation errors (Supplementary Fig. 13a). In conventional tracking approaches using only 2D videos, it is often difficult to correctly track identities when interacting mice are closely interacting, allo-grooming, or passing over and under each other. Although swapped identities can be corrected later once the mice are well-separated again, this still leaves individual behavior during the actual social interaction unresolved[5,16]. We found that our tracking algorithm was robust against both identity swaps (Supplementary Fig. 13b–e) and body direction swaps (Supplementary Fig. 14). This observation agrees with the fact that tracking in 3D space (subject to our implemented spatiotemporal regularizations) should in principle allow better identity tracking. In full 3D space it is easier to determine who is rearing over whom during an interaction, for example.

To test our algorithm for subtler errors, we manually inspected 500 frames, randomly selected across an example 21 min recording session. In these 500 frames, we detected only a single tracking mistake, corresponding to 99.8% correct tracking (Supplementary Fig. 15a). The identified tracking mistake was visible as a large, transient increase in the point-cloud loss function (Supplementary Fig. 15b). After the tracking mistake, the robust particle filter quickly recovered to correct tracking again (Supplementary Fig. 15c). By detecting such loss function anomalies, or by detecting 'unphysical' postures or movements in the body models, potential tracking mistakes can be automatically 'flagged' for inspection (Supplementary Fig. 15c, d). After inspection, errors can be manually corrected or automatically corrected in many cases, for example by tracking the particle filter backwards in time after it has recovered. As the algorithm recovers after a tracking mistake, it is generally unnecessary to actively supervise the algorithm during tracking, and manual inspection for potential errors can be performed after running the algorithm overnight. We provide a GUI for viewing and quality control of tracked behavior (raw data, body skeleton, ellipsoid surfaces, and time trajectory) running in an interactive Jupyter notebook (Supplementary Fig. 2b and Supplementary Video 5).

**Automated analysis of movement kinematics and social events**. As a validation of our tracking method, we demonstrate that our methods can automatically extract both movement kinematics and behavioral states (movement patterns, social events) during spontaneous social interactions. Some unsupervised methods for discovering structure and states in behavioral data do not rely on an explicit body model of the animal, and instead, use statistical methods to detect behavioral states directly from tracked features[6,33,51–53]. In an alternative approach, some supervised methods label behavioral events of interest by hand on training

data, and then train a classifier to find similar events in unlabeled data[17–19]. Both of these types of analysis are compatible with our method (e.g., by running directly on the time series data of the 17 dimensions that parametrize the body models of the two animals, Supplementary Fig. 11). Our tracking system yields an easily interpretable 3D body model of the animals, which makes two additional types of analyses straightforward as well: First, we can easily define 3D body postures or multi-animal postures of interest as templates[16,30]. Second, we can use unsupervised methods to discover behavioral states in the 3D reference frame of the animal's own body, making these models and states straightforward to interpret and 'sanity check' (manually inspect for errors).

To demonstrate posture-template-based analysis, we defined social behaviors of interest as templates and matched these templates to tracked data. We know that anogenital sniffing[54] and nose-to-nose touch[55] are prominent events in rodent social behavior, so we designed a template to detect these events. In this template, we exploited the fact that we could easily calculate both body postures and movement kinematics, in the reference frame of each animal's own body. For every frame, we first extracted the 3D coordinates of the body model skeleton (Supplementary Fig. 12a). From these skeleton coordinates, we calculated the position (Fig. 5a) and a three-dimensional speed vector for each mouse ('forward speed', along the hip ellipsoid, 'left speed' perpendicular to the body axis, and 'up speed' along the $z$-axis; Fig. 5b). We also calculated three instantaneous 'social distances', defined as the 3D distance between the tip of each animal's noses ('nose-to-nose'; Fig. 5b), and from the tip of each animal's nose to the posterior end of the conspecific's hip ellipsoid ('nose-to-tail'; Fig. 5b). From these social distances, we could automatically detect when the mouse bodies were in a nose-to-nose or a nose-to-tail configuration (Fig. 5c). It is straightforward to further subdivide these social events by body postures and kinematics, in order to separate stationary nose-to-tail configurations (anogenital sniffing/grooming) and nose-to-tail configurations during locomotion (social following).

To demonstrate unsupervised behavioral state discovery, we used GPU-accelerated probabilistic programming[56] and state space modeling to automatically detect and label movement states. To discover types locomotor behavior, we fitted a 'sticky' multivariate hidden Markov model[57] to the two components of the speed vector that lie in the $xy$-plane (Supplementary Fig. 16a–h). With five hidden states, this model yielded interpretable movement patterns that correspond to known mouse locomotor 'syllables': resting (no movement), turning left and right, and moving forward at slow and fast speeds (Fig. 5d). Fitting a similar model with three hidden states to the $z$-component of the speed vector (Supplementary Fig. 16i–n) yielded interpretable and known 'rearing syllables': rest, rearing up, and ducking down (Fig. 5e). Using the maximum *a posteriori* probability from these fitted models, we could automatically generate locomotor ethograms and rearing ethograms for the two mice (Fig. 5b).

In line with previous observations, we found that movement bouts were short (medians, rest/left/right/fwd/fast-forward: 0.83/0.50/0.52/0.45/0.68 s, a 'sub-second' timescale[33]). In the locomotion ethograms, bouts of rest were longer than bouts of movement (all $p < 0.05$, Mann–Whitney U-test; Fig. 5f) and bouts of fast forward locomotion was longer than other types of locomotion (all $p < 0.001$, Mann–Whitney U-test; Fig. 5f). In the rearing ethograms, the distribution of rests was very wide, consisting of both long (~seconds) and very short (~tenths of a second) periods of rest (Fig. 5g). As expected, by plotting the rearing height against the duration of rearing syllables, we found that short rests in rearing were associated with standing high on the hind legs (the mouse rears up, waits for a brief moment before ducking back down), while longer rests happened when the mouse was on the ground ('rearing' and 'crouching', Fig. 5h). Like the movement types and durations, the transition probabilities from the fitted hidden Markov models were also in agreement with known behavioral patterns. In the locomotion model, for example, the most likely transition from "rest" was to "slow forward". From "slow forward", the mouse was likely to transition to "turning left", "fast forward" or "turning right", it was unlikely to transition directly from "fast forward" to "rest" or from "turning left" to "turning right, and so on (Supplementary Fig. 16o, p).

**Automatic measurement of firing rate modulations during social touch.** By combining our tracking system with silicon-probe recording of single unit activity, we could automatically measure how neural activity is modulated during social interactions. As proof-of-concept for our system, we implanted a male mouse with a 32-channel silicon probe electrode in barrel cortex (the primary whisker representation in somatosensory cortex). In an example experiment, we simultaneously recorded 31 single units in barrel cortex while tracking the behavior of the implanted mouse interacting with a male and a female conspecific for 20 min each. We then used the posture-template-based analysis to detect three types of social touch events: nose-to-nose touch ("Nose ↔ Nose"), the implanted animal touching the partner's anogenital region with its whiskers ("Nose0 → Tail1") and the partner animal touching the implanted animal's anogenital region with its whiskers ("Nose1 → Tail0", Fig. 6a). The automatic posture-template-based analysis confirmed[58] that the duration of social touch events and inter-touch-intervals spanned multiple orders of magnitude (from short millisecond touch events to longer touch events lasting multiple seconds, Fig. 6b–d).

Using a 'classic' peri-stimulus time histogram-based analysis, we found several single units that had a significant firing modulation at the time of the detected social touch events (example neurons shown in Fig. 6e, top row, labeled "naïve PSTH"). The firing rate modulations detected in the "naïve" approach were surprisingly small (only a small 'bump' in the PSTH at the time of touch), and much smaller than observed in 'classic' barrel cortex studies, where a controlled whisker stimulus is presented[59]. We wondered if the magnitude of firing rate modulation appeared small in the PSTHs, because during untrained and self-initiated behavior, the detected touch events occurred in close temporal proximity and/or were overlapping with other touch events and postural changes[58]. To test the possibility that larger effects sizes were masked by other touch events occurring in close temporal proximity, we also computed PSTHs where we only included social touch events where no other social touch event was detected in the 'baseline' period (4 s before the social touch). In these PSTHs with a "cleaned" baseline (Fig. 6e, bottom row, labeled "cleaned PSTH"), we both observed a larger proportion of neurons with a significant change in firing rate (Fig. 6f) and a larger effect size compared to the naïve PSTHs (Fig. 6g, the distributions of effect sizes in the cleaned PSTH are "wider"). For example, the third neuron shown in Fig. 6e showed no firing rate modulation in the naïve PSTH, but instead showed a large, highly statistically significant firing rate decrease around whisker touch in the "cleaned" PSTH.

**Fully automatic mapping of 'social receptive fields'.** Cleaning the PSTHs (by controlling for only three types of social touch) increased our estimates of the magnitude of firing rate modulations associated with social touch events. However, a PSTH-based analysis strategy has inherent drawbacks when analyzing

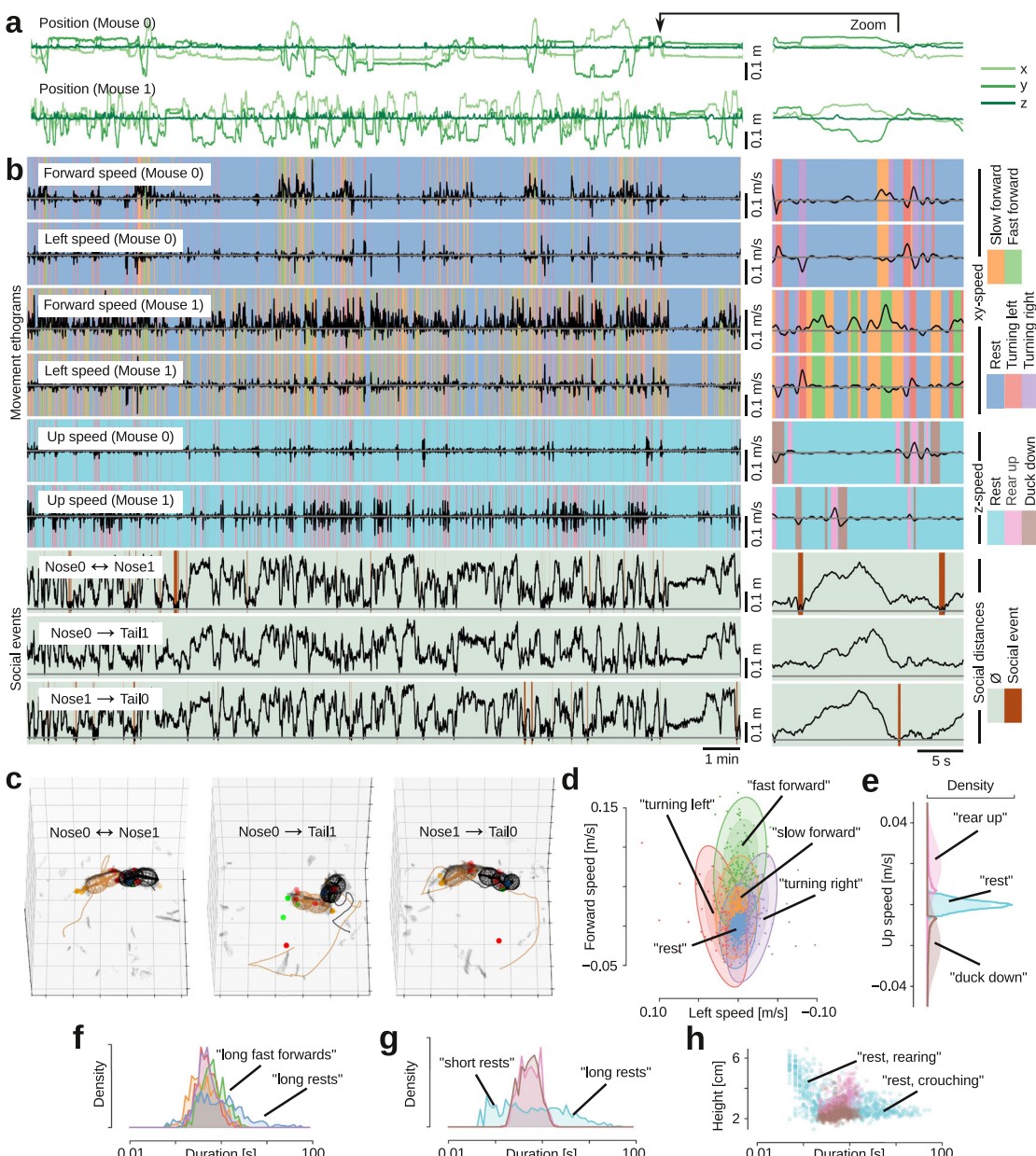

**Fig. 5 Automatic classification of movement patterns and behavioral states during social interactions. a** Tracked position of both mice, across an example 21 min recording. **b** Extracted behavioral features: three speed components (forward, left, and up in the mice's egocentric reference frames), and three 'social distances' (nose-to-nose distance and two nose-to-tail distances). Colors indicate ethograms of automatically detected behavioral states. **c** Examples of identified social events: nose-to-nose-touch, and anogenital nose-contacts. **d** Mean and covariance (3 standard deviations indicated by ellipsoids) for each latent state for the forward/leftward running (dots indicate a subsample of tracked speeds, colored by their most likely latent state) **e** Mean and variance of latent states in the z-plane (shaded color) as well as distribution of tracked data assigned to those latent states (histograms) **f** Distribution of the duration of the five behavioral states in the xy-plane. Periods of rest (blue) are the longest ($p < 0.05$, two-sided Mann–Whitney U-tests) and bouts of fast forward movement (green) are longer other movement bouts ($p < 0.001$, two-sided Mann–Whitney U-tests). **g** Distribution of duration of the three behavioral states in the z-plane. Periods of rest (light blue) are either very short or very long. **h** Plot of body elevation against behavior duration. Short periods of rest happen when the z-coordinate is high (the mouse rears up, waits for a brief moment before ducking back down), whereas long periods of rest happen when the z-coordinate is low (when the mouse is resting or moving around the arena, $\rho = -0.47$, $p < 0.001$, two-sided Spearman's rank correlation coefficient test). Source data and analysis scripts that generate the figures are available in the associated code/data files (see "Methods").

naturalistic behavior. During free behavior, touch, movement, and postural changes happen simultaneously, as continuous and overlapping variables. Furthermore, in line with "vicarious" somatosensory responses reported in human somatosensory cortex[60] and barrel cortex responses observed just before touch[61], barrel cortex neurons may be related to the behavior of the partner animal, in a kind of "mirror neuron"-like response.

To deal with these challenges, we drew inspiration from the discovery of multiplexed spatial coding in hippocampal circuits[62] and developed a fully-automatic python pipeline that can automatically discover 'social' receptive fields. Our tracking method is able to recover the 3D posture and head direction of both animals: The head direction of the implanted animal was given by the skeleton of the body model (the implant is fixed to

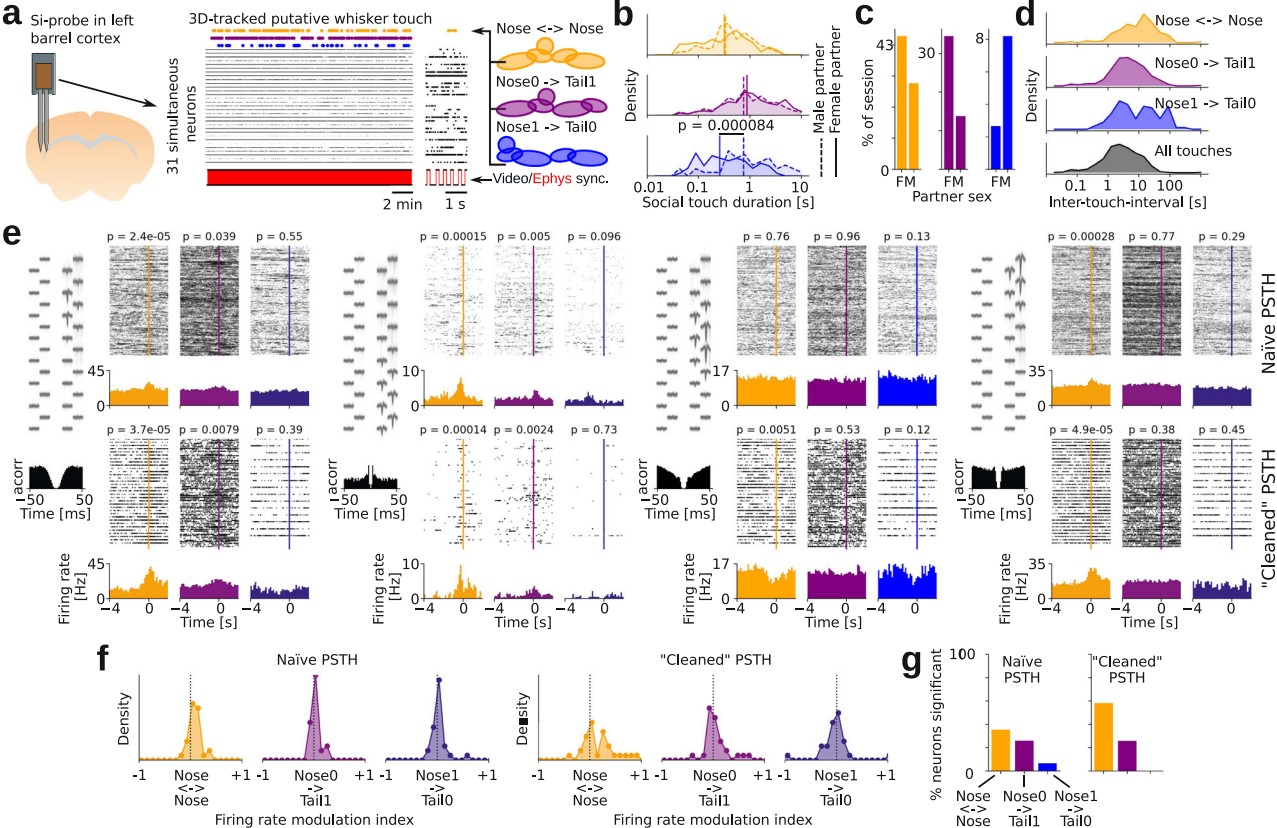

**Fig. 6 Automatic measurement of firing rate modulations during social touch. a** Automatically-detected social touch events in mouse implanted with silicon probe (Si-probe) with 31 single-units from barrel cortex during a single 20 min behavioral session. Yellow, nose-to-nose; purple, implanted-nose-to-partner-tail; blue, partner-nose-to-implanted-tail. **b** Distribution of touch durations with male (dashed) and female (solid) partner ($p = 0.000084$, $N = 64/46$ male/female, two-sided Mann–Whitney U-test). **c** Percentage of behavioral session classified as social touch events, by partner sex, for two behavioral sessions. **d** Distribution of inter-touch-intervals for the two example behavioral sessions. **e** Social touch PSTHs for four neurons. For each neuron, the top row shows 'naïve' PSTHs (aligned to social touch event) and the bottom row shows 'cleaned' PSTHs (we only include events where no other social touch event occurred in the −4 to 0 s period before the detected social touch). The PSTHs in the bottom row have fewer trials, but show much larger effect sizes. (p-values indicate paired, two-tailed Wilcoxon signed rank tests, see "Methods") **f**, Percentage of neurons that pass a $p < 0.05$ significance criterion, based on the 'naïve' and 'cleaned' PSTHs shown above. **g** Distributions of effect size (measured as a firing rate modulation index), based on the 'naïve' and 'cleaned' PSTHs shown above. Source data and analysis scripts that generate the figures are available in the associated code/data files (see "Methods").

the head). For computational efficiency, we exploited the rotational symmetry of the body model of the non-implanted partner to decrease the dimensionality of the search space during tracking (Fig. 4c) and used the 3D coordinates of the detected 'ear' key-points to infer the 3D head direction of the partner (Supplementary Figs. 17, 18).

Using the full 3D body model of both animals, we designed our analysis pipeline to automatically extract 45 continuous features that might be associated with firing rate changes in a social interaction: social "between-animal" features (nose-to-nose distance, nose-to-partner's-genitals distance, relative orientation of the partner with respect to the implanted animal, and a temporal derivative of the distance between the center of the two hip ellipsoids that measures if the animals are moving towards each other or away from each other, Fig. 7a), postural features (head yaw/pitch/roll, etc.), spatial features (to detect 'spatial' activity, such as place fields, border or head-direction activity), movement features (temporal derivatives of the running trajectory, temporal derivatives of posture angles, etc.), and posture, space and movement features of the partner animal (Fig. 7b, Supplementary Fig. 19a, detailed feature table in "Methods").

We assumed the following generative model of the observed neuronal spike trains[62]: A neuron's spike train is generated by a Poisson process, and the rate of this Poisson process is

determined by a linear combination of the behavioral predictors: Each predictor is associated with its own tuning curve, all tuning contributions are summed and passed through an exponential nonlinearity to map the rate of the Poisson process to a positive value (Fig. 7c). To determine what behavioral features significantly contribute to the firing rate modulation of a neuron, and the associated tuning curves, we used a model comparison approach: Starting from a null model where the observed spikes are simply generated by a Poisson process with a constant rate, we iteratively added predictors that passed a cross-validated significance criterion (a significant increase in likelihood compared to a simpler model). The tuning curves were regularized to be smooth and allowed to be re-fit with each additional predictor added to the multiplexed code (details in "Methods").

Using this analysis approach, we found several neurons with a multiplexed encoding of features of the social interactions (Fig. 7d, e). Because of the 3D body models, the discovered neural coding schemes were straightforward to interpret and compare to expected touch-related response patterns in barrel cortex[59]. For example, the example neuron shown in Fig. 7d is strongly modulated by social facial touch (strongly tuned to a low nose-to-nose distance) and strongly lateralized (the neuron is strongly tuned to orientation angle, with a peak at ~ $-\pi/2$, i.e.,

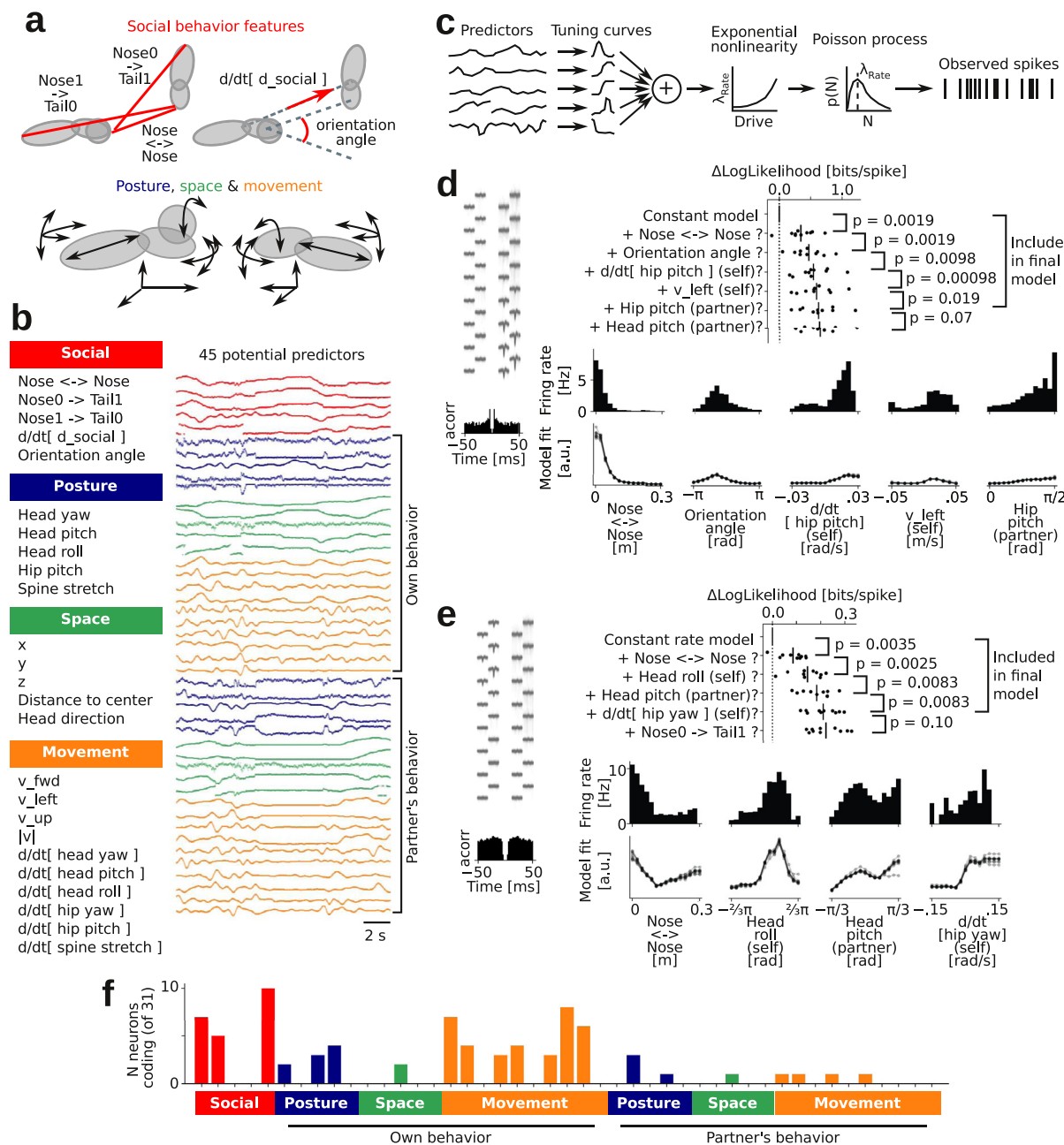

**Fig. 7 Automatic mapping of neural receptive fields in a natural 'social situation'. a** Schematic depiction of automatically extracted social features (top: nose-to-nose and nose-to-tail distances, center-to-center velocity, and head-center-to-head-center angle) and movement/posture features (bottom: rotation and movement of the body model ellipsoids). **b** Names and example traces of extracted behavioral features: social features (red color) and movement (yellow), posture (blue), and spatial (green) features, for both the subject and partner animal. **c** Schematic depiction of the generative model: We assume that every behavioral feature ('predictor') is associated with a tuning curve and that spikes are generated by a Poisson process. **d** Model selection history and associated p-values of each included predictor for an example neuron (average spike shape and ISI-histogram shown to the left, p-values indicate a one-sided, paired cross-validation test, $N = 10$, see "Methods"). The 'raw' marginal firing rate distribution (bars), and the fitted multiplexed tuning curves (10 lines, one line for each data fold) of the identified predictors are shown below. The barrel cortex neuron multiplexes five features, including nose-to-nose distance (the neuron fires more when this is close to zero, i.e., when noses touch) and orientation angle (the neuron fires most at roughly $-\pi/2$, i.e., when the partner is on the right side, the contralateral side relative to the recording electrode). **e** Another example neuron (same plotting and statistical tests as in panel **d**). This barrel cortex neuron multiplexes four features: during nose-to-nose touch, when turning or rolling the head to the right, when partner's nose is tilted up, or when partner's nose is slightly downwards. **f** Distribution of the number of neurons that encode the tested behavioral features (ordering as in **b**). The neurons mainly encoded social touch features (nose-to-nose, implanted-nose-to-partner-tail, and orientation angle) and movement/posture features of the implanted animal itself (blue and yellow bars, above 'own behavior'). Source data and analysis scripts that generate the figures are available in the associated code/data files (see "Methods").

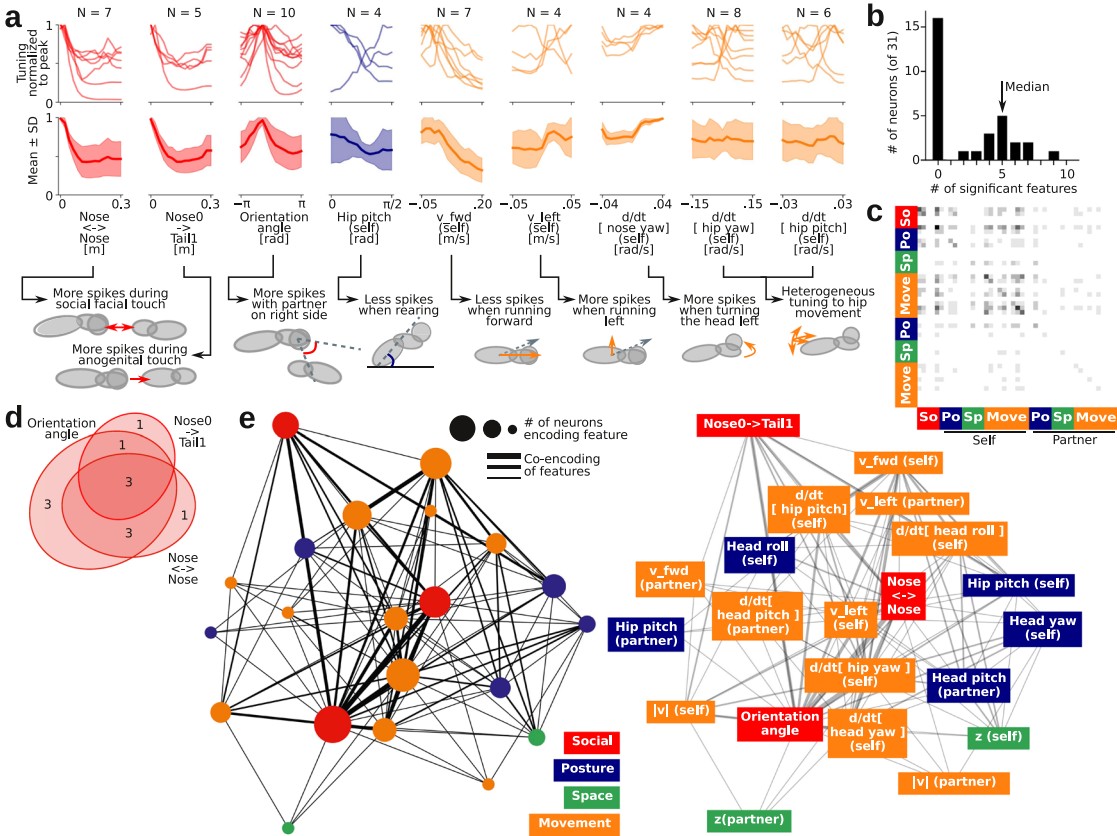

**Fig. 8 Population tuning and co-encoding network structure in a social situation. a** Top, single neuron tuning curves and 'population tuning curve' (average tuning, shaded area indicates standard deviation) for all behavioral features encoded by more than three single neurons. Bottom, schematic depiction of the physical interpretation of the population tuning, in relation to the 3D body models. **b** Distribution of the number of behavioral features that each single neuron multiplexes. The arrow indicated the median number of features encoded by a neuron that encode at least one feature. **c** Co-encoding matrix of the neural population: The grayscale color in i'th and j'th bin in the heatmap indicates the number neurons that encode both feature i and j (ordering and color on the axes as in Fig. 7). **d** Euler diagram of a subset of the co-encoding matrix: This shows the number of neurons that encode nose-to-nose touch, implanted-nose-to-partner-tail touch, and orientation angle (i.e., are lateralized). **e** Network graph depiction of the full co-encoding matrix. The size of the nodes indicates the number of neurons that encode a feature, the width of the edges indicates the number of neurons that co-encode a behavioral feature. The network is shown in the Kamada–Kawai projection[86] (the distance between nodes approximate their graph-theoretical distance), with additional text labels on the network on the right. Source data and analysis scripts that generate the figures are available in the associated code/data files (see "Methods").

when the partner is on the contralateral side of the animal's face, relative to the implanted recording electrode). The example neuron shown in Fig. 7e was also strongly tuned to social facial touch (tuned to a low nose-to-nose distance), was strongly tuned to a positive head roll (i.e., when the head is turned such that the whisker field contralateral to the recording electrode is in contact with the floor) and was strongly tuned to a positive temporal derivative of the hip ellipsoid yaw (when the animal is running counterclockwise, e.g., along the edge of the arena, such that the contralateral whisker field is brushing against the arena wall or other obstacles). Across the population, we found that the neurons overwhelmingly encoded whisker touch and orientation angle (lateralization), and the posture and movements of the implanted animal, but not the partner animal (Fig. 7f).

**Mapping the network topology of social responses.** To map how neurons across the population might also be tuned to features of social interactions, we extracted the estimated neural tuning curves of all features that were encoded by at least 4 neurons (Fig. 8a). For some features, there was a clear pattern across the population, in line with known response patterns in barrel cortex[59]: All neurons that were modulated by social touch increased their firing rate during touch (tuned to a low nose-to-

nose and nose-to-tail distance), were tuned to touch contralateral to the implanted electrode (tuning peak at orientation angle ≈ $-\pi/2$), and decreased firing rate during higher locomotion speeds (negatively correlated with forward speed). For the remaining movement and posture features, the tuning was more heterogeneous across the population (Fig. 8a).

Finally, our automatic tracking and tuning curve estimation pipeline makes it straightforward to determine how features might be multiplexed together in the same neurons. In our example session, we found that 52% of the neurons encoded at least one behavioral feature, with a median number of five encoded features (Fig. 8b). Using all neurons that encoded at least one feature, we computed a population "co-encoding matrix", where the entries of the matrix are the probability that two features are encoded by the same neuron (Fig. 8c and Supplementary Fig. 19b). This co-encoding matrix was structured, such that there was a large overlap between neurons that encode nose-to-nose touch, neurons that encode nose-to-partner-genital touch, and neurons that had lateralized responses (modulated by the relative orientation angle of the animals, preferring touch to the contralateral whisker field, relative to the implanted recording electrode[59], Fig. 8d). The co-encoding matrix specified a network graph of encoded features (Fig. 8e),

which would then be amenable to various methods of network topology analysis (e.g., locality, clustering, subgraph motifs, etc.). Thus, our fully-automatic pipeline enables direct connections from raw behavioral videography and spike train recordings to higher-order statistics about how features of a social interaction are mapped onto a neural population during naturalistic behavior.

## Discussion

We combined 3D videography, deep learning, and GPU-accelerated robust optimization to estimate the posture dynamics of multiple freely-moving mice, engaging in naturalistic social interactions. Our method is cost-effective (requiring only inexpensive consumer depth cameras and a GPU), has high spatio-temporal precision, is compatible with neural implants for continuous electrophysiological recordings, and tracks unmarked animals of the same coat color (e.g., enabling behavioral studies in transgenic mice). Our method is fully automatic, which makes the method scalable across multiple PCs or GPUs. Automatic tracking allows us to investigate social behavior across long behavioral time scales beyond what is feasible with manual annotation, for example to elucidate developmental trajectories, dynamics of social learning, or individual differences among animals[63,64], among other types of questions. Finally, our method uses no message-passing from future frames, but only relies on past data, which makes the method a promising starting point for real-time tracking. A major next step for future work is to apply such algorithms to animal behavior in different conditions. For example, the algorithm can easily be adapted to track other animal body shapes such as juvenile mice, other species, or movable, deformable objects that are part of more complex experimental environments.

In social interactions, rodents respond to the behavior of conspecifics, but we are only beginning to discover how the rodent brain encodes complex features such as gaze direction or body postures of others[3,21,65,66]. Compared to our knowledge about, for example, sensorimotor mirror neurons in monkeys[67] and vicarious sensory responses in human subjects[60] (both foundational to theories about human social cognition[68] and empathy[69]), we still know very little about a putative rodent mirror neuron system[69]. For demonstration and validation, we applied our analysis pipeline to barrel cortex neurons, and were able to recover expected neural tuning to (lateralized) whisker touch and movement[59]. Our end-to-end tracking method and analysis pipeline maps tuning to movements and postures of the partner's body, and is thus ideally suited to detect potential social interaction systems such as rodent 'mirror neuron' signals in other brain areas[70,71]. The 45 predictor features that we have included in our analysis pipeline could be expanded to add additional features of interest. Similar to multiplexed spatial tuning in parahippocampal cortices (e.g., "conjunctive" grid- and head-direction cells[72]), we model multiplexed tuning as multiplicative[62]. It is straightforward to modify our model comparison code to also consider other coding schemes, such as nonlinear or conditional interactions between predictors. This is of particular interest to the social neuroscience of joint action, where movements and postures can have particular social meaning when performed in coordination with a social partner[21].

Social dysfunctions can be devastating symptoms in a multitude of mental conditions, including autism spectrum disorders, social anxiety, depression, and personality disorders[73]. Social interactions also powerfully impact somatic physiology, and social interactions are emerging as a promising protective and therapeutic element in somatic conditions, such as inflammation[74] and chronic pain[75]. Disorders characterized by deficits in social interaction and

communication generally lack effective treatment options, largely because even the neurobiological basis of 'healthy' social behavior is poorly understood. In addition to relating behavior to neural activity, automated 3D body tracking can yield a high-fidelity readout of behavioral changes associated with manipulations of neural activity, both at short (e.g., optogenetic), medium (e.g., pharmacological), and long (e.g., gene knockout) time scales.

Long-term multi-animal behavior tracking has a particular advantage in comparative social neuroscience. For example, human genomics have linked several genes to autism[2–4], but we still know little about *how* these genetic changes increase the risk of autism. A 'computational ethology'[76] approach to social behavior analysis based on automatic posture tracking (such as pioneered in laboratory studies of insects, worms, and fish[20,77–82] and in field ethology[83–86]) does not require us to a priori imagine how, e.g., autism-related gene perturbations manifest in mice, but can identify subtle changes in higher-order behavioral statistics without human observer bias. By recording long periods of social interactions, it may be possible to use methods from computational topology to ask how the high-dimensional space defined by touch, posture, and movement dynamics is impacted by different genotypes or pathological conditions. The statistical power and granularity of the long-term continuous 3D behavior data may allow us to identify what specific core components of social behaviors are altered in different social relations, by various neuroactive drugs, and in disease states[53].

Our algorithm is automatic, does not use any message-passing from future frames, and robustly recovers from tracking mistakes. Thus, it is possible in principle to run the algorithm in real-time. Currently, the processing time per frame is higher than the camera frame rate, but the algorithm is also not yet fully optimized for speed. For example, in the current version of the algorithm, we first record the images to disk, and then read and pre-process the images later. This is convenient for algorithm development and exploration, but writing and reading the images to disk, and moving them onto and off a GPU are time-intensive steps. Going forward, it is important to explore ways to increase tracking robustness further, such as for example using the optical flow between video frames to link key-points together in multi-animal tracking[15], using a 3D convolutional neural network to detect body key-points by considering 'un-projected' views from all cameras around the behavioral arena simultaneously[10], real-time painting-in of depth artifacts[87], and better online trajectory forecasting with a network trained to propose trajectories based on previously tracked mouse movements. Experimentation and optimization are clearly needed, but our algorithm—requiring data transfer from only a few cameras, with deep convolutional networks, physical modeling, and particle filter tracking implemented as tensor algebra on the same GPU—is a promising starting point for the development of real-time, multi-animal 3D tracking, compatible with head-mounted electrophysiology.

## Methods

**Animal welfare and ethics**. All experimental procedures were performed according to animal welfare laws under the supervision of local ethics committees. All procedures were approved under NYU School of Medicine IACUC protocols. Animals were kept on a 12h/12h light cycle with ad libitum access to food and water. Mice presented as partner animals were housed socially in same-sex cages, and post-surgery implanted animals were housed in single animal cages. Neural recordings electrodes were implanted on the dorsal skull under isoflurane anesthesia, with a 3D-printed electrode drive and a hand-built mesh housing.

**Hardware**. Necessary hardware and approximate prices are shown in Table 1. Setting up the system also requires general lab electronics (tape, wire, soldering equipment, etc.), including: • Four infrared or red LEDs. • Two 0.1" pin headers or jumper wires. • Four 20 kOhm resistors. • Four 22 nF capacitors. • One 200 Ohm resistor (or same order of magnitude). • One stick (for moving ping-pong ball during calibration).

**Table 1 Necessary hardware.**

| Item | Recommendation | Price (USD) | N | Total (USD) |
|---|---|---|---|---|
| Depth cameras | Intel RealSense D435 | 179.00 | 4 | 716.00 |
| Camera stands | Etubby 26″ gooseneck webcam stand | 24.96 | 4 | 99.84 |
| PCIe card with 4 independent USB 3.0 controllers | Startech 4-port superspeed | 83.54 | 1 | 83.54 |
| Active, repeating USB 3.0 cables | UGREEN, USB 3.0 Active Repeater Cable | 18.89 | 4 | 75.56 |
| Arduino with USB cable | Arduino Uno R3 | 13.98 | 1 | 13.98 |
| Pytorch-compatible GPU | Any NVIDIA card with CUDA support | 500.00 | 1 | 500.00 |
| Behavioral arena (acrylic cylinder or elevated platform) | 12″-diameter, 5/32″ thick acrylic cylinder | 71.20 | 1 | 71.20 |
| Depth camera GPIO pin connector (jumper) | JST ASSHSSH28K305 | 0.54 | 8 | 4.32 |
| Depth camera GPIO pin connector (jumper housing) | JST SHR-09V-S | 0.19 | 4 | 0.76 |
| Colored ping-pong balls (for calibration) | Stiga 40 mm ITTF Regulation size | 6.64 | 1 | 6.64 |
| Total | | | | 1571.84 |

**Software**. Our system uses the following software: Linux (tested on Ubuntu 16.04 LTE, but should work on others, https://ubuntu.com/), Intel Realsense SDK (https://github.com/IntelRealSense/librealsense), Python (tested on Python 3.6). Required Python packages will be installed with PIP or conda (script in supplementary software). All required software is free and open source.

**Computer hardware**. All experiments and benchmarks were done on a desktop PC running Ubuntu 16.04 LTE on a 3.7 GHz 6-core CPU (Intel i7-8700K), with 32 GB 2400 MHz RAM, and an Nvidia GeForce RTX 2080Ti GPU.

**Recording data structure**. The Python program is set to pull raw images at 640 × 480 (color) and 640 × 480 (depth), but only saves 320 × 210 (color) and 320 × 240 (depth). We do this to reduce noise (multi-pixel averaging), save disk space and reduce processing time. Our software also works for saving images up to 848 × 480 (color) and 848 × 480 (depth) at 60 frames/s, in case the system is to be used for a bigger arena, or to detect smaller body parts (e.g., eyes, paws). Images were transferred from the cameras with the python bindings for the Intel Realsense SDK v. 2.0 (https://github.com/IntelRealSense/librealsense) and saved as 8-bit, 3-channel PNG files with opencv (for color images) or as 16-bit binary files (for depth images).

**3D data structure**. For efficient access and storage of the large datasets, we save all pre-processed data to hdf5 files. Because the number of data points (point-cloud and key-points) per frame varies, we save every frame as a jagged array. To this end, we pack all pre-processed data to a single array. If we detect N points in the point-cloud and M key-points in the color images, we save a stack of the 3D coordinates of the points in the point-cloud (Nx3, raveled to 3N), the weights (N), the 3D coordinates of the key-points (Mx3, raveled to 3M), their pseudo-posterior (M), an index indicating key-point type (M), and the number of key-points (1). Functions to pack and unpack the pre-processed data from a single line ('pack_-to_jagged' and 'unpack_from_jagged') are provided.

**Temporal synchronization**. LED blinks were generated with voltage pulses from an Arduino (on digital pin 12), controlled over USB with a python interface for the Firmata protocol (https://github.com/tino/pyFirmata). To receive the Firmata messages, the Arduino was flashed with the 'StandardFirmata' example, which comes with the standard Arduino IDE. TTL pulses were 150 ms long and spaced by ~$U(150,350)$ ms. We recorded the emitted voltage pulses in both the infrared images (used to calculate the depth image) and on a TTL input on an Open Ephys Acquisition System (https://open-ephys.org/). We detected LED blinks and TTL flips by threshold crossing and roughly aligned the two signals by the first detected blink/flip. We first refined the alignment by cross correlation in 10 ms steps, and then identified pairs of blinks/flips by detecting the closest blink, subject to a cutoff (zscore < 2, compared to all blink-to-flip time differences) to remove blinks missed by the camera (because an experimenter moved an arm in front of a camera to place a mouse in the arena, for example). The final shift and drift were estimated by a robust regression (Theil-Sen estimator) on the pairs of blinks/links.

**Deep neural network**. We used a stacked hourglass network[34] implemented in Pytorch[44] (https://github.com/pytorch/pytorch). The network architecture code is from the implementation in 'PyTorch-Pose' (https://github.com/bearpaw/pytorch-pose). The full network architecture is shown in Supplementary Fig. 1. The Image augmentation during training was done with the 'imgaug' library (https://github.com/aleju/imgaug). Our augmentation pipeline is shown in Supplementary Figure 3. The 'fake laser dot pattern' was generated using the 'snowflakes' generator in the imgaug routines for generating weather effects, tuned to look—by eye—to a similar dot size and density to the real laser dot pattern. The network was trained by RMSProp ($\alpha = 0.99$, $\varepsilon = 10^{-8}$) with an initial learning rate of 0.00025. During training, the learning rate was automatically reduced by a factor of 10 if the training

loss decreased by less than 0.1% for five successive steps (using the built-in learning rate scheduler in Pytorch). After training, we used the final output map of the network for key-point detection, and used a maximum filter to detect key-point locations as local maxima in network output images with a posterior pseudo-probability of at least 0.5.

**Image labeling and target maps**. For training the network to recognize body parts, we need to generate labeled frames by manual annotation. For each frame, 1–5 body parts are labeled on the implanted animal and 1–4 body parts on the partner animal. This can be done with any annotation software; we used a modified version of the free 'DeepPoseKit-Annotator'[8] (https://github.com/jgraving/DeepPoseKit-Annotator/) included in the Supplementary Code. This software allows easy labeling of the necessary points, and pre-packages training data for use in our training pipeline. Body parts are indexed by i/p for implanted/partner animal ('nose_p' is the nose of the partner animal, for example). Target maps were generated by adding a Gaussian function ($\sigma = 3$ px for implant, $\sigma = 1$ px for other body parts, scaled to peak value = 1) to an array of zeros (at 1/4th the resolution of the input color image) at the location of every labeled body key-point. 1D part affinity maps were created by connecting labeled key-points in an array of zeros with a 1 px wide line (clipped to max value = 1), and blurring the resulting image with a Gaussian filter ($\sigma = 3$ px).

**Aligning depth and color data**. The camera intrinsics (focal lengths, $f$, optical centers, $p$, depth scale, $d_{scale}$) and extrinsics (rotation matrices, R, translation vectors, $\bar{t}$) for both the color and depth sensors can be accessed over the SDK. Depth and color images were aligned to each other using a pinhole camera model. For example, the $z$ coordinate of a single depth pixel with indices ($i_c$, $i_d$) and 16-bit depth value, $d_{ij}$, is given by:

$$z_d = d_{ij} \cdot d_{scale} \tag{1}$$

And the $x$ and $y$ coordinates are given by:

$$\begin{bmatrix} x_d \\ y_d \end{bmatrix} = \begin{matrix} (j_d - p_{x,d}) \cdot z_d/f_{x,d} \\ (i_d - p_{y,d}) \cdot z_d/f_{y,d} \end{matrix} \tag{2}$$

Using the extrinsics between the depth and color sensors, we can move the coordinate to the reference frame of the color sensor:

$$\begin{bmatrix} x \\ y \\ z \end{bmatrix}_c = R_{d \to c} \begin{bmatrix} x \\ y \\ z \end{bmatrix}_d + \bar{t}_{d \to c} \tag{3}$$

Using the focal length and optical center, we can project the pixel onto the color image:

$$\begin{bmatrix} i_c \\ j_c \end{bmatrix} = \begin{matrix} f_{y,c} \cdot y_c/z_c + p_{y,c} \\ f_{x,c} \cdot x_c/z_c + p_{x,c} \end{matrix} \tag{4}$$

For assigning color pixel values to depth pixels, we simply rounded the color pixel indices ($i_c$, $i_d$) to the nearest integer and cloned the value. More computationally intensive methods based on ray-tracing exist ('rs2_project_color_pixel_to_depth_pixel' in the Librealsense SDK, for example), but the simple pinhole camera approximation yielded good results (small jitter average out across multiple key-points) which allowed us to skip the substantial computational overhead of ray tracing for our data pre-processing.

**Depth camera calibration, exposure, and 3D alignment**. To align the cameras in space, we first mounted a blue ping-pong ball on a stick and moved it around the behavioral arena while recording both color and depth video. For each camera, we used a combination of motion filtering, color filtering, smoothing, and thresholding to detect the location of the ping-pong ball in the color frame (details in code). We

then aligned the color frames to depth frames and extracted the corresponding depth pixels, yielding a partial 3D surface of the ping-pong ball. By fitting a sphere to this partial surface, we could estimate the 3D coordinate of the center of the ping-pong ball (Fig. 3a). This procedure yielded a 3D trajectory of the ping-pong ball in the reference frame of each camera (Fig. 3b). We used a robust regression method (RANSAC routines to fit a sphere with a fixed radius of 40 mm, modified from routines in https://github.com/daavoo/pyntcloud), insensitive to errors in the calibration ball trajectory to estimate the transformation matrices needed to bring all trajectories into the same frame of reference (Fig. 3c). The software includes a step-by-step recipe for performing the alignment procedure. The depth cameras have a minimum working distance of 20 cm, so they must be placed at least this distance from the behavioral arena. The depth map is calculated from the infrared camera stream, so—as with the RGB video—it is important that the image is not under- or over-exposed. The code includes a tool for streaming live video from all cameras to verify that: (i) the whole arena is in view of all the cameras and (ii) that the exposure is reasonable. The exposure settings can be changes in the config files, that are loaded and applied when recording (the Intel RealSense SDK demo C application library also includes a nice tool for testing different exposure settings). The 3D pixel density drops off with distance from the camera (following the inverse-square law). In our tested use (standard neuroscience behavioral arena, max. ~ 1 × 1 m), the exact relative placement of the four depth cameras does not matter (as they are aligned by the calibration). However, for very large arenas, it may be necessary to add more depth cameras (additional cameras mounted above the arena, for example). Adding more cameras will only affect the pre-processing time (can be run in parallel—which can minimize the impact of more cameras), not the actual body model fitting time (the slowest part of the algorithm). The body model fitting time is determined by the number of mice tracked (the particle filter sorting step scales exponentially with the number of mice, because the algorithm evaluates multi-animal poses).

**Body model**. We model each mouse at two prolate ellipsoids. The model is specified by the 3D coordinate of the center of the hip ellipsoid, $\bar{c}_{hip} = [x, y, z]$, and the major and minor axis of the ellipsoids are scaled by a coordinate, $s \in [0, 1]$ that can morph the ellipsoid from long and narrow to short and fat:

$$a_{hip} = a_{hip,0} + a_{hip,\Delta} \cdot s \tag{5}$$

$$b_{hip} = b_{hip,0} + b_{hip,\Delta} \cdot (1 - s) \tag{6}$$

The 'neck' (the joint of rotation between the hip and nose ellipsoid) is sitting a distance, $d_{hip} = 0.75 \cdot a_{hip}$, along the central axis of the hip ellipsoid. In the frame of reference of the mouse body (taking $\bar{c}_{hip}$ as the origin, with the major axis of the hip ellipsoid along the $x$-axis), a unit vector pointing to of the nose ellipsoid, from the 'neck' to the center of the nose ellipsoid along the major axis is:

$$\bar{e}_{nose} = \begin{bmatrix} \cos\theta \\ \sin\theta\cos\phi \\ \sin\theta\sin\phi \end{bmatrix} \tag{7}$$

In the frame of reference of the laboratory ('world coordinates'), we allow the hip ellipsoid to rotate around the $z$-axis ('left'/'right') and around the $y$-axis ('up'/'down', in the frame of reference of the mouse). We define $\mathbf{R}(\alpha_x, \alpha_y, \alpha_z)$ as a 3D rotation matrix specifying the rotation by an angle $\alpha$ around the three axes, and $\mathbf{R}(\bar{v}_1, \bar{v}_2)$ as a 3D rotation matrix that rotates the vector $\bar{v}_1$ onto $\bar{v}_2$. The we can define:

$$\mathbf{R}_{hip} = \mathbf{R}(0, \beta, \gamma) \tag{8}$$

$$\mathbf{R}_{head} = \mathbf{R}(\bar{e}_x, \bar{e}_{nose}) \tag{9}$$

where $\bar{e}_x$ is a unit vector along the $x$-axis. In the frame of reference of the mouse body, the center of the nose ellipsoid is:

$$\bar{c}_{nose,mouse} = \mathbf{R}_{head}\begin{bmatrix} d_{nose} \\ 0 \\ 0 \end{bmatrix} + \begin{bmatrix} d_{hip} \\ 0 \\ 0 \end{bmatrix} \tag{10}$$

So, in world coordinates, the center is:

$$\bar{c}_{nose,world} = \mathbf{R}_{hip}c_{nose,mouse} + \bar{c}_{hip} \tag{11}$$

The center of the neural implant if offset from the center of the nose ellipsoid by a distance $x_{impl}$ along the major axis of the nose ellipsoid, and a distance $z_{impl}$ orthogonal to the major axis. We allow the implant to rotate around the nose ellipsoid by an angle, $\psi$. Thus, in the frame of reference of the mouse body, the center of the ellipsoid is:

$$\bar{c}_{impl,mouse} = \mathbf{R}_{head}\begin{bmatrix} s_{impl} \\ d_{impl} \cdot \cos\psi \\ d_{impl} \cdot \sin\psi \end{bmatrix} + \begin{bmatrix} d_{hip} \\ 0 \\ 0 \end{bmatrix} \tag{12}$$

And in world coordinates, same as the center of the nose ellipsoid:

$$\bar{c}_{impl,world} = R_{hip}c_{impl,mouse} + \bar{c}_{hip} \tag{13}$$

We calculated other skeleton points (tip of the nose ellipsoid, etc.) in a similar method. We can use the rotation matrices for the hip and the nose ellipsoids to calculate the characteristic ellipsoid matrices:

$$\mathbf{Q}_{hip} = \mathbf{R}_{hip}\begin{bmatrix} 1/a_{hip}^2 & 0 & 0 \\ 0 & 1/b_{hip}^2 & 0 \\ 0 & 0 & 1/b_{hip}^2 \end{bmatrix}(\mathbf{R}_{hip})^T \tag{14}$$

$$\mathbf{Q}_{nose} = \mathbf{R}_{hip}\mathbf{R}_{head}\begin{bmatrix} 1/a_{nose}^2 & 0 & 0 \\ 0 & 1/b_{nose}^2 & 0 \\ 0 & 0 & 1/b_{nose}^2 \end{bmatrix}(\mathbf{R}_{hip}\mathbf{R}_{head})^T \tag{15}$$

Calculating the shortest distance from a point to the surface of an 3D ellipsoid in 3 dimensions requires solving a computationally-expensive polynomial[39]. Doing this for each of the thousands of points in the point-cloud, multiplied by four body ellipsoids, multiplied by 200 particles pr. fitting step is not computationally tractable. Instead, we use the shortest distance to the surface, $\tilde{d}$, along a path that passes through the centroid (Supplementary Fig. 8a, b). This is a good approximation to d (especially when averaged over many points), and the calculation of $\tilde{d}$ can be implemented as pure vectorized linear algebra, which can be calculated very efficiently on GPU[40]. Specifically, to calculate the distance from any point $\bar{p}$ in the point-cloud, we just center the points on the center of an ellipsoid, and—for example—calculate:

$$\bar{p}' = \bar{p} - \bar{C}_{hip} \tag{16}$$

$$\tilde{d} = |1 - \|\bar{p}'\|_{hip}^{-1}| \cdot \|\bar{p}'\| \tag{17}$$

where, $\|\bar{p}'\|_{Q_{hip}} = \sqrt{\langle\bar{p}', \bar{p}'\rangle_{Q_{hip}}} = \sqrt{(\bar{p}')^T Q_{hip}\bar{p}'} \tag{18}$

In fitting the model, we used the following constants: $a_{nose} = 2.00$ cm, $b_{nose} = 1.20$ cm, $a_{hip(min)} = 0.50$ cm, $a_{hip(max)} = 2.50$ cm, $b_{hip(min)} = 1.20$ cm, $b_{hip(max)} = 1.50$ cm, $d_{nose} = 1.00$ cm, $d_{hip} = 0.75 \cdot a_{hip}$, $r_{impl} = 0.9 \cdot b_{nose}$, $x_{impl} = d_{nose} + 0.5 \cdot a_{nose}$, $z_{impl} = 1.5 \cdot r_{impl}$. The code includes a parameter ('body_scale') that can be changed to scale the mouse body model (e.g., for other strains, or juvenile mice).

**Loss function evaluation and tracking**. Joint position of the two mice is represented as a particle in 17-dimensional space. For each data frame, we start with a proposal particle (leftmost green block, based on previous frames), from which we generate 200 particles by pseudo-random perturbation within a search space (next green block). For each proposal particle, we calculate three types of weighted loss contributions: loss associated with the distance from the point-cloud to the surface of the mouse body models (top path, green color), loss associated with body key-points (middle path, key-point colors as in and loss associated with overlap of the two mouse body models (bottom path, purple color). We broadcast the results in a way, which allows us to consider all $200 \times 200 = 40.000$ possible joint postures of the two mice. After calculation, we pick the top 200 joint postures with the lowest overall loss, and anneal the search space, or—if converged—continue to the next frame. When we continue to a new frame, we add the fitted frame to an online recursive filter bank, which proposes the next position of the particle for the next frame, based on previous frame. All loss function calculations, and recursive filter predictions are implemented as pure tensor algebra, fully vectorized, and executed on a GPU.

**Online recursive filtering**. To propose a new location for the particle filter between frames, we use a recursive least squares filter[89], with a time embedding of 5 steps, a forgetting factor of $\mu = 0.99$ and a regularization factor of $\varepsilon = 0.1$. Our implementation ('rls_bank') is based on the implementation in the Padasip (Python Adaptive Signal Processing) library (https://github.com/matousc89/padasip). For the first 150 frames, the filter is only trained, but after frame 150, the filter is used for prediction. The code allows this filter to run across all dimensions of the particle filter, but—in practical use—we found it sufficient to run it across the $x$-, $y$- and $z$- coordinates of the center of the two mouse body models (i.e., we just assume that the angular and stretch coordinates do not change from the last frame—this saves a few computations, and can be selected by commenting in/out the relevant lines in the code).

**Regularizations**. To regularize the particle filter algorithm, we imposed two hard rules ('barriers') on the movement of the body models (shown in Supplementary Fig. 8). The first barrier was implemented by adding a large term to the particle filter's loss function, if the center of any ellipsoids from two different bodies were closer than 0.8 times the sum of their short axes (this barrier allows a 20% overlap of spheres with a radius equal to the ellipsoid's small axis, drawn in purple in Supplementary Fig. 8f). This barrier term prevents 'unphysical' overlaps between the body models of the two mice. The second barrier was implemented by adding a

large term to the particle filter's loss function, if the same condition was met between the current position of a mouse body model and the interaction partners body model *in the preceding frame* (Supplementary Fig. 8h). This barrier term prevents 'flips' between the two mice (where the body models change identity), as drawn in Supplementary Fig. 8g.

**State space filtering of raw tracking data.** After tracking, the coordinates of the skeleton points ($c_{hip}$, $c_{nose}$, etc.) were smoothed with a 3D kinematic Kalman filter tracking both the 3D position ($p$), velocity ($v$), and (constant) acceleration ($a$). For example, for the center of the hip coordinate:

$$\bar{x} = \left[ p_x, v_x, a_x, p_y, v_y, a_y, p_z, v_z, a_z \right] \tag{19}$$

$$\bar{z} = \left[ c_{hip,x}, c_{hip,y}, c_{hip,z} \right] \tag{20}$$

$$\mathbf{F} = \begin{bmatrix} \mathbf{F}' & \mathbf{0} & \mathbf{0} \\ \mathbf{0} & \mathbf{F}' & \mathbf{0} \\ \mathbf{0} & \mathbf{0} & \mathbf{F}' \end{bmatrix}, \text{where } \mathbf{F}' = \begin{bmatrix} 1 & dt & \frac{1}{2}dt^2 \\ 0 & 1 & dt \\ 0 & 0 & 1 \end{bmatrix} \tag{21}$$

$$\mathbf{H} = \begin{bmatrix} 1 & 0 & 0 & 0 & 0 & 0 & 0 & 0 & 0 \\ 0 & 0 & 0 & 1 & 0 & 0 & 0 & 0 & 0 \\ 0 & 0 & 0 & 0 & 0 & 0 & 1 & 0 & 0 \end{bmatrix} \tag{22}$$

$$\mathbf{P} = \mathbf{1}_{9 \times 9} \cdot \sigma_{cov}^2 \tag{23}$$

$$\mathbf{R} = \mathbf{I}_{3 \times 3} \cdot \sigma_{measurement}^2 \tag{24}$$

$$\mathbf{Q} = \begin{bmatrix} \mathbf{Q}' & \mathbf{0} & \mathbf{0} \\ \mathbf{0} & \mathbf{Q}' & \mathbf{0} \\ \mathbf{0} & \mathbf{0} & \mathbf{Q}' \end{bmatrix} \cdot \sigma_{process}^2 \tag{25}$$

where Q' is the Q matrix for a discrete constant white noise model $\mathbf{Q}' =$

$\begin{bmatrix} \frac{1}{4}dt^4 & \frac{1}{2}dt^3 & \frac{1}{2}dt^2 \\ \frac{1}{2}dt^3 & dt^2 & dt \\ \frac{1}{2}dt^2 & dt & 1 \end{bmatrix}$ and $\sigma_{measurement} = 0.015$ m, $\sigma_{process} = 0.01$ m, $\sigma_{cov}^2 =$ 0.0011 m². The $\sigma$'s were the same for all points, except the slightly more noisy estimate of the center of the implant, where we used. $\sigma_{measurement} = 0.02$ m, $\sigma_{process} = 0.01$ m, $\sigma_{cov}^2 = 0.0011$ m² From the frame rate (60 fps), $dt = \frac{1}{60}$ s. The maximum-likelihood trajectory was estimated with the Rauch–Tung–Striebel method[48] with a fixed lag of 16 frames. The filter and smoother was implemented using the 'filterpy' package (https://github.com/rlabbe/filterpy). The spine scaling, s, was smoothed with a similar filter in 1D, except that we did not model acceleration, only s and a (constant) s 'velocity', with $\sigma_{measurement} = 0.3$, $\sigma_{process} = 0.05$ m, $\sigma_{cov}^2 = 0.0011$.

After filtering the trajectories of the skeleton points, we recalculated the 3D rotation matrices of the hip and head ellipsoid by the vectors pointing from $c_{hip}$ to $c_{mid}$ (from the middle of the hip ellipsoid to the neck joint), and from $c_{hip}$ to $c_{nose}$ (from the neck joint to the middle of the nose ellipsoid). We then converted the 3D rotation matrixes to unit quaternions, and smoothed the 3D rotations by smoothing the quaternions with an 10-frame boxcar filter, essentially averaging the quaternions by finding the largest eigenvalue of a matrix composed of the quaternions within the boxcar[50]. After smoothing the ellipsoid rotations, we re-calculated the coordinates of the tip of the nose ellipsoid ($c_{tip}$) and the posterior end of the hip ellipsoid ($c_{tail}$) from the smoothed central coordinates, rotations, and—for $c_{tail}$—the smoothed spine scaling. A walkthrough of the state space filtering pipeline is shown in Supplementary Fig. 12.

**Template matching.** To detect social events, we calculated three social distances, from three instantaneous 'social distances', defined as the 3D distance between the tip of each animal's noses ('nose-to-nose'), and from the tip of each animal's nose to the posterior end of the conspecific's hip ellipsoid ('nose-to-tail'; Fig. 5c). From these social distances, we could automatically detect when the mouse bodies were in a nose-to-nose (if the nose-to-nose distance was < 2 cm and the nose-to-tail distance was > 6 cm) and in a nose-to-tail configuration (if the nose-to-nose distance was > 6 cm and the nose-to-tail distance was > 2 cm). The events were detected by the logic conditions, and then single threshold crossings due to noise were removed by binary opening with a 3-frame kernel, followed by binary closing with a 30-frame kernel.

**State space modeling of mouse behavior.** State space modeling of the locomotion behavior was performed in Pyro[56] a GPU-accelerated probabilistic programming language built on top of Pytorch[44]. We modeled the (centered and whitened) locomotion behavior as a hidden Markov model with discrete latent states, z, and

associated transition matrix, **T**.

$$z(t + 1) = \text{Categorical}(e_{z(t)}^T \cdot \mathbf{T}) \tag{26}$$

$$\mathbf{T} = \begin{bmatrix} p_{ij} & \cdots \\ \vdots & \ddots \end{bmatrix} \tag{27}$$

To make the model 'sticky' (discourage fast switching between latent states) we draw the transition probabilities, $p_{ij}$ from a Dirichlet prior with a high mass near the 'edges' and initialize $\mathbf{T}_{init} = (1 - \eta)\mathbf{I} + \eta/n_{states}$ where $\eta = 0.05$.

$$p \sim \text{Dirichlet}(0.5) \tag{28}$$

Each state emits a forward speed and a left speed, drawn from a two-dimensional Gaussian distribution with a full covariance matrix.

$$\begin{bmatrix} v_{fwd} \\ v_{left} \end{bmatrix} \sim \text{MVNormal}(\mu, \mathbf{S}) \tag{29}$$

We draw the mean of the states from a normal distribution and use a LKJ Cholesky prior for the covariance:

$$\mu \sim \text{Normal}(0, 1) \tag{30}$$

$$\mathbf{S} = \begin{bmatrix} \sigma_{fwd} & 0 \\ 0 & \sigma_{left} \end{bmatrix} \mathbf{L} \begin{bmatrix} \sigma_{fwd} & 0 \\ 0 & \sigma_{left} \end{bmatrix} \tag{31}$$

$$\sigma \sim \text{LogNormal}(-1, 1) \tag{32}$$

$$\mathbf{L} \sim \text{LKJcorr}(2) \tag{33}$$

The up speed was modeled in a similar way, except that the latent states were just a one-dimensional normal distribution. The means and variances for the latent states were initialized by kmeans clustering of the locomotion speeds. The model was fit in parallel to 600-frame snippets of a subset of the data by stochastic variational inference[90]. We used an automatic delta guide function ('AutoDelta') and an evidence lower bound (ELBO) loss function. The model was fitted by stochastic gradient descent with a learning rate of 0.0005. After model fitting, we generated the ethograms by assigning latent states by maximum a posteriori probability with a Viterbi algorithm.

**3D head direction estimation.** We use the 3D position of the ear key-points to determine the 3d head direction of the partner animal. We assign the ear key-points to a mouse body model by calculating the distance from each key-point to the center of the nose ellipsoid of both animals (cutoff: closest to one mouse and < 3 cm from the center of the head ellipsoid, Supplementary Fig 17a). To estimate the 3D head direction, we calculate the unit rejection ($v_{rej}$) between a unit vector along the nose ellipsoid ($v_{nose}$) and a unit vector from the neck joint ($c_{mid}$) to the average 3D position of the ear key-points that are associated with that mouse (v_ear_-direction, Supplementary Fig. 17b). If no ear key-points were detected in a frame, we linearly interpolate the average 3D position. To average out jitter, the estimates of the average ear coordinates and the center of the nose coordinate were smoothed with a Gaussian ($\sigma = 3$ frames). The final head direction vector was also smoothed with a Gaussian ($\sigma = 10$ frames).

**Extracellular recording and spike clustering.** Extracellular recordings were made with sharpened 2-shank, 32-site NeuroNexus P2 profile silicon probes (Neuro-Nexus Technologies, Inc., MI, USA). The silicon probes were implanted in barrel cortex using a stereotax (1 mm posterior, 3.2 mm lateral to bregma[91]) under isoflurane anesthesia using a custom 3D printed plastic microdrive and base plates for mice, shielded by a copper mesh and bound to the animal's skull using dental cement[92]. The neural data was recorded using an Intan RHD 32-channel headstage with accelerometer (Intan Technologies, CA, USA) connected to an Open Ephys Acquisition Board[93] (https://open-ephys.org/) at 30 kHz/16 bit resolution. The neural data was pre-clustered using SpyKING CIRCUS[94] (a custom probe geometry file for the P2 probe and the full clustering script with all parameters is available in the supplementary code) and checked manually for cluster quality in KLUSTA[95]. Only well-separated single units were included in the analysis.

**PSTH-based analysis of neural responses.** For the PSTH-based analysis, we triggered on the three social events detected as described under 'Template matching'. For the 'naïve' PSTH, we included all events, and for the 'cleaned' PSTH, we only included events, where there were no other of the detected events occurring in the preceding 4 s. Significant firing rate changes were detected by comparing the average firing rate, $r_{pre}$, between −4 and −2 s (relative to the start of the detected event) with the average firing rate, $r_{post}$, between −0.5 and 0.5 s, using a Wilcoxon signed rank test, at $p < 0.05$. The firing rate modulation index was

calculated using the same firing rates and defined as:

$$\text{Mod.idx.} = \frac{r_{\text{post}} - r_{\text{pre}}}{r_{\text{post}} + r_{\text{pre}}} \tag{34}$$

**Statistical modeling of neural tuning curves**. Our spike train modeling approach is based on ref. [62] and our python code for model fitting and model selection is based on the supplementary Matlab code from that study (available at https://github.com/GiocomoLab/ln-model-of-mec-neurons). We calculated multiple features of the 'social scene', describing the social, postural, spatial, and movement dynamics (definitions, bin ranges, tuning curve boundary conditions, etc. are detailed in Supplementary Table 1). In the table, we only list the variables associated with the posture, spatial location, and movement of the implanted animal (subscript 0). We include identical features for the partner animal (subscript 1). The bin rages were selected to span the physically possible values (e.g., within the circular arena), or to span the observed values in the behavior (for movement speeds, for example).

We model the observed spike train as generated by the following process (Fig. 7c): The spikes are generated by a Poisson process. The rate of the Poisson process is determined by the features, in the following way: Each feature is multiplied with a tuning curve (taking any real value), to generate a weight. The weights of all features are summed, pass through an exponential nonlinearity (to clamp the rate of the Poisson process to be positive). This means that in the spike rate space, the tuning to the features is multiplicative.

We convert each feature into binary dummy variables by binning (bins listed in the table above) to generate a time-by-bins matrix, $A$, where the $i$'th and $j$'th index is a binary variable indicating if the feature was in the $j$'th feature bin in the $i$'th frame. If we let $\bar{c}$ be a column vector with the values of the tuning curve for a single predictor, then our linear model says that the rate of the Poisson process generating the spikes, $\lambda$, depending on $p$ predictors can be expressed as

$$\bar{\lambda} = \exp\left(\sum_p A_p \bar{c}_p\right)/dt \tag{35}$$

We fit the linear model by tuning the parameters of the tuning curves to maximize of the Poisson log-likelihood of the observed number of spikes, $n$, in each bin of the spike train. We include a regularization term, $\beta$, that ensures that the tuning curves are smooth (it is a loss term associated with the difference between $c_i$ and $c_{i+1}$, with circular wrap-around for the circular features). Thus, the fitted tuning curves are:

$$\hat{c} = \text{argmax}_c \sum_i \log P\left(n_i|\exp\left(\sum_p A_p \bar{c}_p\right)\right) - \sum_p \beta\left(\sum_i \frac{1}{2}\left(c_{p,i} - c_{p,i+1}\right)^2\right) \tag{36}$$

We fit the models using the Newton conjugate gradient trust-region algorithm ('trust-ncg' method in 'minimize' in the SciPy optimize module, using the Taylor expansion approximation to the Jacobian and Hessian and a tolerance of 1e−3).

To determine which features significantly contribute to the firing rate modulation of a neuron, we use a cross-validated model comparison approach, and a greedy forward selection of features. First, we compare a fitted 'baseline' model where the spikes are simply generated by a Poisson process with a constant rate to 45 fitted models, that include only one feature. The comparison is cross-validated, such that we fit the model on 90% of the data and evaluate on 10% held-out data (with 3 skips, i.e., we split the data in 30 chunks, fit to 27, and evaluate on 3). To compare each of the one-feature models to the baseline model, we calculate the increase in log-likelihood of the test data, given the fitted one-feature models (relative to the baseline model), across all 10 permutations of the 10-fold cross validation. We select the best candidate feature (defined as the one with the highest average increase in log-likelihood, across the 10 folds), and check if the increase in log-likelihood is significant by performing a one-sided Wilcoxon signed-rank test, with a criterion of $p < 0.05$. If the best candidate feature is significant, we add that feature to a library of features that we consider significant for that neuron. If we have the number of spikes in the spike train, $\bar{n}$, and the maximum-likelihood fitted rate is $\bar{\lambda}(\hat{c})$, then the log-likelihood increase, $\Delta\mathcal{L}$ (in bits/spike) is:

$$\mathcal{L}_{\text{model}} = \left(\sum_i \lambda_i - n_i \log(\lambda_i) + \log(n_i!)\right)/ \tag{37}$$

$$\mathcal{L}_{\text{constant}} = \left(\sum_i \langle n\rangle - n_i \log(\langle n\rangle) + \log(n_i!)\right)/\sum_i n_i \tag{38}$$

$$\Delta\mathcal{L} = -\log(2) \cdot \left(\mathcal{L}_{\text{model}} - \mathcal{L}_{\text{constant}}\right) \tag{39}$$

For all $(N > 1)$-feature models (two features, three features, etc.), we use the same approach: We fit all possible models that add one more feature to the library of $N − 1$ significant features (all tuning curves of all features in the library are re-fit every time), we select the best candidate feature, and use a one-sided Wilcoxon signed-rank test between a model with $N$ features and a model with $N − 1$ features to determine if that candidate feature is significant and should be added to the library. If the one-sided Wilcoxon signed-rank test is not significant at $p < 0.05$, we stop the search for new features to add to the library.

**Population structure analysis**. The Euler diagram in Fig. 8d was drawn in R using the eulerr package[96]. The network co-encoding graph shown in Fig. 8e was drawn in the Kamada–Kawai projection[88] (the distance between nodes approximate their graph-theoretical distance), using the NetworkX python package[97].

**Reporting summary**. Further information on research design is available in the Nature Research Reporting Summary linked to this article.

## Data availability

Source data (and an example dataset and raw electrophysiology data) is available as a Source Data File (*.xlsx files) and/or on Zenodo: https://doi.org/10.5281/zenodo.5790820. The raw high-speed video files are available upon request to the corresponding author, requests will be answered within 2 weeks. Source data are provided with this paper.

## Code availability

Code and instructions for data recording (python functions) and data analysis (python functions and Jupyter notebooks) are available on Github: https://github.com/chrelli/3DDD_social_mouse_tracker/.

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

## Acknowledgements
This work was supported by The Novo Nordisk Foundation (C.L.E.), the NIH (DC012557, HD088411, and NS107616 to R.C.F.), and a Howard Hughes Medical Institute Faculty Scholarship (R.C.F.). We thank György Buzsáki, David Tingley, and Manuel Valero for help in establishing silicon probe recordings and for the gift of 3D printed silicon drive parts.

## Author contributions
C.L.E. designed and implemented the system, performed experiments, analyzed the data, made figures, and wrote the first version of the manuscript. R.C.F. supervised the study. C.L.E and R.C.F. wrote the manuscript.

## Competing interests
The authors declare no competing interests.
