## [Peer Review File · Nature Communications]

Reviewers' Comments:

Reviewer #2:

Remarks to the Author:

Thank you for the detailed responses to my initial review (Reviewer 2). My concerns have been satisfactorily addressed with the inclusion of your quantitative results in Supp. Fig 4-7. (You could say more in the main text about the surprising performance hit you take by adding PAFs and why you think they are still advantageous.)

The inclusion of the social modeling and the corresponding neural encoding strengthens the paper, makes it more relevant to the neuroscience audience, and distinguishes this work from other recent papers.

With these changes and additions, I think this paper is well suited for publication in Nature Communications.

Reviewer #3:

Remarks to the Author:

Overall the manuscript has improved significantly in revision – the authors have implemented several major improvements to the manuscript:

1) They have toned down overstatements about the broad utility of their method, and removed text that was confusing or misleading.

2) They have now made more systematic comparisons (and included text in the Discussion) with other available methods, including LMT and MoSeq, as well as SimBA and MARS – I am convinced by the authors arguments about the comparisons and satisfied with the text they have added. The authors now better explain the value of the 3D body model their method generates, relative to other methods.

3) The authors have added new data/figures of proof of principle experiments showing “social receptive field” analysis, in which they record from neurons while tracking behavior and develop a model for relating the two types of data. This is very nice and a welcome addition to the paper (Reviewer Figure 2.2 is especially helpful as it makes clearer the potential value of the 3D body model).

4) The authors have added Supplementary Figures to quantify network performance – this is very important for assessing the accuracy, sensitivity, and flexibility of their method.

5) The authors have added new experiments to address concerns about the flexibility of their system with regard to variation in lighting conditions

6) Additionally with regard to comments from other Reviewers, there was a concern about the use of PAFs and their utility here – Supp Fig 7 is useful, but the authors could also cite the SLEAP study (Pereira et al.) that more systematically explores different network configurations and pipelines (including the use of PAFs) for keypoint tracking.

The concerns with generalizability of the method remain, but are mitigated by the transfer to Nature Communications and the addition of new figures to show the utility of the method for correlating neural activity and social behavior.

I don't have any additional suggestions or concerns, and congratulate the authors on a job well done.

Point-by-point response to the reviewers

Reviewer #2 (Remarks to the Author):

Thank you for the detailed responses to my initial review (Reviewer 2). My concerns have been satisfactorily addressed with the inclusion of your quantitative results in Supp. Fig 4-7. (You could say more in the main text about the surprising performance hit you take by adding PAFs and why you think they are still advantageous.)

We agree that we could expand and add more discussion on the PAFs. However, it is a somewhat tangential point to the (already very long) manuscript. In order to keep the main text focused, we prefer to keep the discussion in the supplementary material as it is now.

The inclusion of the social modeling and the corresponding neural encoding strengthens the paper, makes it more relevant to the neuroscience audience, and distinguishes this work from other recent papers.

With these changes and additions, I think this paper is well suited for publication in Nature Communications.

We thank the referee for the constructive comments and thoughtful suggestions on our manuscript.

Reviewer #3 (Remarks to the Author):

Overall the manuscript has improved significantly in revision ??? the authors have implemented several major improvements to the manuscript:

- 1) They have toned down overstatements about the broad utility of their method, and removed text that was confusing or misleading.*
- 2) They have now made more systematic comparisons (and included text in the Discussion) with other available methods, including LMT and MoSeq, as well as SimBA and MARS ??? I am convinced by the authors arguments about the comparisons and satisfied with the text they have added. The authors now better explain the value of the 3D body model their method generates, relative to other methods.*
- 3) The authors have added new data/figures of proof of principle experiments showing ???social receptive field??? analysis, in which they record from neurons while tracking behavior and develop a model for relating the two types of data. This is very nice and a welcome addition to the paper (Reviewer Figure 2.2 is especially helpful as it makes clearer the potential value of the 3D body model).*
- 4) The authors have added Supplementary Figures to quantify network performance ??? this is very important for assessing the accuracy, sensitivity, and flexibility of their method.*
- 5) The authors have added new experiments to address concerns about the flexibility of their system with regard to variation in lighting conditions*
- 6) Additionally with regard to comments from other Reviewers, there was a concern about the use of PAFs and their utility here ??? Supp Fig 7 is useful, but the authors could also cite the SLEAP study (Pereira et al.) that more systematically explores different network configurations and pipelines (including the use of PAFs) for keypoint tracking.*

We agree with this suggestion and we have added a citation to SLEAP (Pereira et al.) for the use of PAFs (ref. 15 on line 178).

The concerns with generalizability of the method remain, but are mitigated by the transfer to Nature Communications and the addition of new figures to show the utility of the method for correlating neural activity and social behavior.

I don't have any additional suggestions or concerns, and congratulate the authors on a job well done.

We thank the referee for the constructive and useful suggestion on our manuscript.